

# Correlation coefficients on normal wiggly dual hesitant fuzzy sets: an application in the selection of real estate agents

Kamran Kausar[1], Khizar Hayat[1], Dragan Pamucar[2] and Nadeem Ajaib[1]

[1] Department of Mathematics, University of Kotli, AJ&K, Kotli, Azad Kashmir, Pakistan
[2] Széchenyi István University, Gyor, Hungary

## ABSTRACT

Decision makers (DMs) continually demonstrate shortcomings in their approaches to analyzing information through fuzzy systems; nevertheless, a model that integrates many dimensions of uncertainty is generally substantial. Normal wiggly dual hesitant fuzzy sets (NWDHFSs) incorporate a range of DMs' preferences for membership grades (MGs) and non-membership grades (NMGs). For complicated and multifaceted problems, one can apply the dynamic framework of NWDHFSs. To illustrate the relationship between NWDHFSs, correlation coefficients (CCs) on NWDHFSs, as well as weighted CCs on NWDHFSs, are presented in this work. These CCs are built up using means of values in hesitant fuzzy elements of NWDHFSs. Some fundamental axioms and thresholds of CCs on NWDHFSs are examined. A multi-criteria decision-making (MCDM) technique and associated algorithms based on these CCs are introduced. Because of the competitive real estate market, choosing a real estate agent is a challenging task for organizations. Through the consideration of a real estate case study, we select an appropriate real estate agent for a real estate firm utilizing proposed CCs on NWDHFSs. We examine the methodologies and outcomes of our approach to previous strategies.

# INTRODUCTION

A correlation coefficient (CC) is a statistical measure that demonstrate the strength of the association between two variables as well as the direction in which the relationship is changing. It assesses the level to which changes in one variable are dependent with changes in another variable. The fuzzy CCs are extensions of traditional CCs, such as Pearson's CC, it designed to handle data that is imprecise, uncertain, or ambiguous. In traditional statistics, CCs identify the strength and direction of a linear relationship between two variables. However, when information of an object contains fuzziness (*e.g.*, when resultant values are vague or imprecise), traditional correlation frameworks may not be suitable or applicable. The fuzzy CCs are designated to quantify the magnitude of similarity or relationship between fuzzy sets (FSs) (*Zadeh, 1965*). Initially, *Yu (1993)* proposed the concept of CCs on fuzzy numbers. *Chiang & Lin (1999)* researched an alternative approach for CCs on FSs.

Corresponding authors
Khizar Hayat,
khizarhayat@uokajk.edu.pk
Dragan Pamucar,
pamucar.dragan@sze.hu

The MCDM approaches that are used in the modern world make extensive use of CCs to define the relationship between objects, while FSs are utilized to label an enormous parameterized fuzziness in systems. The FSs are insufficient to classify information when multiple credible grades occur in complex systems. *Torra & Narukawa (2009)* expanded conventional FSs by introducing the concept Hesitant Fuzzy Set (HFS) (*Torra, 2010*), which permit the assignment of several believable degrees of uncertain information. Researchers concur that HFSs outshine FSs and are utilized in various domains, including image segmentation, distance analysis (*Xu & Xia, 2011a*), outranking approaches (*Wang et al., 2014*), linguistic decision making (*Lee & Chen, 2015*; *Rodriguez, Martinez & Herrera, 2011*; *Wu, Wang & Wangzhu, 2025*), information measures (*Ajaib et al., 2025*; *Farhadinia, 2013*), aggregation operators (*Mehmood et al., 2019*; *Mehmood et al., 2018*) and pattern recognition (*Luo et al., 2025*; *Qahtan et al., 2025*).

Several features of HFS have been investigated that can be used to build on new ideas in hesitant fuzzy set theory. These include aggregation operators on HFS by *Xia, Xu & Chen (2013)*, entropy and cross-entropy on HFSs by *Xu & Xia (2012)*, distance measures (DMs) and similarity measures (SMs) on HFSs by *Xu & Xia (2011a)*, scores on HF elements (HFEs) (*Alcantud et al., 2023*) and correlation measures on HFSs by *Xu & Xia (2011b)*.

As in elements of HFSs, the information is oscillating between values in the interval $[0, 1]$, it is important to make a better decision on such information; it makes sense to define correlation among HFSs. Various sort of CCs exist concerning HFSs and their utilization throughout multiple disciplines. The essential CCs of HFSs were presented by *Chen, Xu & Xia (2013)*, *Meng & Chen (2015)* and *Liao, Xu & Zeng (2015)*, who utilized them in decision-making and clustering analysis. Probabilistic HFS (PHFS) is another significant type of HFS that takes into account the probability that each membership in HFEs will occur (*Zhang, Xu & He, 2017*; *Zhou & Xu, 2017*).

Given that HFEs in HFSs possess several grades assigned by DMs, a suitable technique was necessary to provide a range of hesitation factors to address the reason of hesitation. To support this motive, *Ren, Xu & Wang (2018)* established the concept of normal wiggly HFSs (NWHFSs) and exploited it for assessments of environmental quality. The NWHFSs consist of normal wiggly elements (NWEs), identified using the statistical concept of standard deviation and the real preference degree proposed by *Yager (1988)*. *Liu et al. (2019)*, *Liu & Zhang (2021)* proposed two categories of aggregation operators for NWHFSs: Muirhead mean operators and Maclaurin mean operators. Subsequently, these operators were employed in MCDM methodologies. *Xia, Chen & Fang (2022)* introduced probabilistic NWHFSs and demonstrated their use in battle threat assessments. *Narayanamoorthy, Ramya & Kang (2020)* and *Wang et al. (2024)* investigated the application of NWHFSs in several aspects of MCDM.

The Atanassov's idea of intuitionistic fuzzy sets (IFSs) (*Atanassov, 1986*) is beneficial when considering NMGs in conjunction with MGs. Inspired by Atanassov's concept, *Zhu & Xu (2014)* and *Zhu, Xu & Xia (2012)* proposed dual HFS (DHFS), wherein both MGs and NMGs encompass value sets from the interval $[0, 1]$. The DHFSs have extended in different version such as, *Alcantud et al. (2019)* consider an additional form of duality factor by proposing dual extended fuzzy sets. *Ye (2013)* has examined the fundamental

CCs on DHFSs and utilized them in MCDM. Following *Ye (2013)*, various foundational aspects of CCs have been demonstrated by researchers (*Tyagi, 2015*). *Meng, Xu & Wang (2019)* defined CCs of DHFSs utilizing DMs for applications in engineering management. *Garg, Sun & Liu (2023)* proposed an algorithm employing CCs on DHFSs for engineering cost management. *Boulaaras et al. (2024)* investigated DMs on DHFSs and application in medical diagnoses. *Ning, Wei & Wei (2024)* introduced probabilistic DHFSs and CCs, while *Karaaslan & Özlü (2020)* explored CCs on type-2 DHFSs and their application in clustering analysis. Furthermore, a robust CCs for probabilistic DHFSs have been investigated by *Garg & Kaur (2020)*.

The concept of NWDHFS is investigated by *Narayanamoorthy et al. (2019)*, it is a contemporary mathematical instrument utilized to describe MGs and NMGs of the uncertain information inherent in the cognitive processes of DMs. Subsequently, *Ali & Naeem (2022)* established DMs and SMs for NWDHFSs and explored its applicability in medical diagnostics. The NWDHFSs consist of duality preferences, hesitancy, and different levels of preference associated with dual hesitant fuzzy elements (DHFE). On the other hand, dual hesitant fuzzy sets have been applied in multi-expert decision-making, Risk assessments and social choice theory. Data's relationship to the best or worse case scenario is crucial when it comes to deep accounting.

Therefore, many scenarios addressing uncertainty in NWDHFSs necessitated a correlation among NWDHFSs to enhance outcomes in MCDM. In light of this purpose, we present CCs on NWDHFSs and weighted CCs on NWDHFSs. This study examines the essential axioms of CCs about NWDHFSs and the interrelations among CCs. We present a multi-criteria decision-making technique and associated algorithms based on these CCs. Through the examination of a real estate case study, we derive appropriate assessments of real estate agents for real estate firms utilizing NWDHFSs. We will analyze the techniques and results of our approach in comparison to several existing techniques.

## PRELIMINARIES

This part will look at fundamental ideas of HFSs and DHFSs, along with their correlation coefficients developed by *Ye (2013)* and *Garg, Sun & Liu (2023)*. Through the assessments of specific instances and properties of DHFSs, we assess the limitations of current computational methods comprehensively. We will review particular concepts of NWDHFSs and identify the distinctions between NWDHFS and DHFS.

The concept of HFS enhanced the notion of FSs, as it encompasses a range of values from the internal interval $[0, 1]$. It is beneficial when a group of prospectors offers evaluations of an object in an uncertain environment. The notion of HFS, proposed by *Torra (2010)*, provides profound insights about its relevance in real-world uncertain circumstances.

**Definition 1 (*Torra, 2010*).** *Let $\mathscr{Z} = \{z_1, z_2, \cdots, z_n\}$ indicates an original fixed universe of discourse. The HFS $\tilde{\mathscr{S}}$ on $\mathscr{Z}$ is expressed as;*

$$\tilde{\mathscr{S}} = \left\{ \langle z_i, \aleph_{\tilde{\mathscr{S}}}(z_i) \rangle \mid z_i \in \mathscr{Z} \right\},$$

where $\aleph_{\tilde{\mathscr{S}}}(z_i)$ represents all possible membership grades from $[0,1]$, it showing hesitancy of certain prospectors toward the object $z_i \in \mathscr{L}$, that is $\aleph_{\tilde{\mathscr{S}}}(z_i) = \underset{\delta \in [0,1]}{\cup} \{\delta\}$. For any $i \in (1,2,\cdots,n)$, $\aleph_{\tilde{\mathscr{S}}}(z_i)$ is known as HFE of HFS $\tilde{\mathscr{S}}$.

*Zhu, Xu & Xia (2012)* provides the definition of DHFSs from the perspective of Atanassov's IFSs (*Atanassov, 1986*). This definition states that both MGs and NMGs are HFEs. It appears that DHFS is a more flexible tool that may be applied in multiple ways DHFS seems to be a more adaptable instrument that may be utilized in many approaches based on current requirements, in contrast to the present FSs and HFSs, while including significantly more information supplied by DMs. DHFS has specific and advantageous characteristics (*Narayanamoorthy et al., 2019*).

**Definition 2** (*Narayanamoorthy et al., 2019*; *Zhu, Xu & Xia, 2012*). *Let* $\mathscr{L} = \{z_1, z_2, \cdots, z_n\}$ *indicates an origional fixed universe of discourse. The DHFS $\tilde{\mathscr{D}}$ on $\mathscr{L}$ is expressed as;*

$$\tilde{\mathscr{D}} = \left\{ \langle z_i, \aleph_{\tilde{\mathscr{D}}}(z_i), \Upsilon_{\tilde{\mathscr{D}}}(z_i) \rangle \mid z_i \in \mathscr{L} \right\},$$

*where* $\aleph_{\tilde{\mathscr{D}}}(z_i) = \underset{\delta \in [0,1]}{\cup} \{\delta\}$ *and* $\Upsilon_{\tilde{\mathscr{D}}}(z_i) = \underset{v \in [0,1]}{\cup} \{v\}$ *represents the membership and non membership grades of $z_i \in \mathscr{L}$ for $\tilde{\mathscr{D}}$, respectively. It holds following axioms;* $0 \le \delta^+ + v^- \le 1$ *and* $0 \le \delta^- + v^+ \le 1$, *where,* $\delta^+ = \underset{\delta \in \aleph_{\tilde{\mathscr{D}}}}{\max}\{\delta\}$, $\delta^- = \underset{\delta \in \aleph_{\tilde{\mathscr{D}}}}{\min}\{\delta\}$,

$v^+ = \underset{v \in \Upsilon_{\tilde{\mathscr{D}}}}{\max}\{v\}$, *and* $v^- = \underset{v \in \Upsilon_{\tilde{\mathscr{D}}}}{\min}\{v\}$. *For any* $z \in \mathscr{L}$, $\langle \aleph_{\tilde{\mathscr{D}}}(z), \Upsilon_{\tilde{\mathscr{D}}}(z) \rangle$ *is called duel hesitant fuzzy element (DHFE).*

In order to investigate the joint relationship and interconnection between two DHFSs sets with the assistance of an interdependence measure, the CC is an efficient instrument to use. *Ye (2013)* is the one who presented the fundamental principle of CC on DHFSs.

**Definition 3** (*Ye, 2013*). *Let* $\mathscr{L} = \{z_1, z_2, \cdots, z_n\}$ *indicates an original fixed universe of discourse. Let* $\tilde{\mathscr{D}} = \left\{ \langle z_i, \aleph_{\tilde{\mathscr{D}}}(z_i), \Upsilon_{\tilde{\mathscr{D}}}(z_i) \rangle \mid z_i \in \mathscr{L} \right\}$ *and* $\tilde{\mathscr{D}}' = \left\{ \langle z_i, \aleph'_{\tilde{\mathscr{D}}}(z_i), \Upsilon'_{\tilde{\mathscr{D}}}(z_i) \rangle \mid z_i \in \mathscr{L} \right\}$. *Then CC on $\tilde{\mathscr{D}}$ and $\tilde{\mathscr{D}}'$ is indicted by*

$$\rho_{DHFS}(\tilde{\mathscr{D}}, \tilde{\mathscr{D}}') = \frac{\sum_{i=1}^{n} \left( \begin{array}{c} \frac{1}{\theta_i} \sum_{k=1}^{\theta_i} \left(\aleph_{\tilde{\mathscr{D}}}\right)_{\varpi(k)}(z_i) \left(\aleph'_{\tilde{\mathscr{D}}}\right)_{\varpi(k)}(z_i) \\ + \frac{1}{\varkappa_i} \sum_{k'=1}^{\varkappa_i} \left(\Upsilon_{\tilde{\mathscr{D}}}\right)_{\varpi(k')}(z_i) \left(\Upsilon'_{\tilde{\mathscr{D}}}\right)_{\varpi(k')}(z_i) \end{array} \right)}{\sqrt{\sum_{i=1}^{n} \left( \begin{array}{c} \frac{1}{\theta_i} \sum_{k=1}^{\theta_i} \left(\aleph_{\tilde{\mathscr{D}}}\right)^2_{\varpi(k)}(z_i) \\ + \frac{1}{\varkappa_i} \sum_{k'=1}^{\varkappa_i} \left(\Upsilon_{\tilde{\mathscr{D}}}\right)^2_{\varpi(k')}(z_i) \end{array} \right)} \sqrt{\sum_{i=1}^{n} \left( \begin{array}{c} \frac{1}{\theta_i} \sum_{k=1}^{\theta_i} \left(\aleph'_{\tilde{\mathscr{D}}}\right)^2_{\varpi(k)}(z_i) \\ + \frac{1}{\varkappa_i} \sum_{k'=1}^{\varkappa_i} \left(\Upsilon'_{\tilde{\mathscr{D}}}\right)^2_{\varpi(k')}(z_i) \end{array} \right)}} \quad (1)$$

*where $\theta_i$ and $\varkappa_i$ are number values in MGs and NMGs of an element $z_i \in \mathscr{L}$ respectively and* $\varpi(k) \ge \varpi(k+1)$ *and* $\varpi(k') \ge \varpi(k'+1)$.

**Example 1** *Let $\tilde{\mathscr{D}}_1, \tilde{\mathscr{D}}_2, \tilde{\mathscr{D}}_3$ and $\tilde{\mathscr{D}}_4$ be four DHFSs;*

$$
\begin{aligned}
\tilde{\mathscr{D}}_1 &= \{\langle z_1, (0.3, 0.19), (0.25, 0.21)\rangle, \langle z_2, (0.4, 0.2), (0.24, 0.22)\rangle\} \\
\tilde{\mathscr{D}}_2 &= \{\langle z_1, (0.45, 0.285), (0.375, 0.315)\rangle, \langle z_2, (0.6, 0.3), (0.36, 0.33)\rangle\} \\
\tilde{\mathscr{D}}_3 &= \{\langle z_1, (0.4, 0.3), (0.26, 0.20)\rangle, \langle z_2, (0.35, 0.22), (0.38, 0.26)\rangle\} \\
\tilde{\mathscr{D}}_4 &= \{\langle z_1, (0.48, 0.36), (0.312, 0.24)\rangle, \langle z_2, (0.42, 0.264), (0.456, 0.31)\rangle\}.
\end{aligned}
$$

*According to the Definition (3), we concluded, $\rho_{DHFS}(\tilde{\mathscr{D}}_1, \tilde{\mathscr{D}}_3) = 0.9759 = \rho_{DHFS}(\tilde{\mathscr{D}}_2, \tilde{\mathscr{D}}_4)$. The CC $\rho_{DHFS}$ (Ye, 2013), have exactly same result for DHFS $\tilde{\mathscr{D}}_1, \tilde{\mathscr{D}}_3$ and $\tilde{\mathscr{D}}_2, \tilde{\mathscr{D}}_4$.*

*One can check that $\tilde{\mathscr{D}}_1$ and $\tilde{\mathscr{D}}_3$ are proportional to $\tilde{\mathscr{D}}_2$ and $\tilde{\mathscr{D}}_4$ respectively. As a result, there is a gap in research that has to be filled in order to design more effective CCs that are able to deal with proportional numbers of DHFSs. A more concise description of CC is required in order to solve such a challenge, hence in this article we will utilize the concept of NWDHFSs for some new CCs.*

In the MCDM process, the degree of hesitation is influenced not just by the format or magnitude of the DM's weighted input but also by the vague and subjective feelings of the DMs. According to constrained logic, an inherent characteristic of humanity, individuals' perceptions of any certain numerical value are likely to exist within a spectrum. The actual configuration of the spectrum may be interval, triangle, trapezoidal, or other forms, which corresponds to the initial assessment data provided by the DMs. To explore the enormous uncertainty of hesitant fuzzy information, *Ren, Xu & Wang (2018)* developed a visualization that identifies the standard oscillatory spectrum for each value in a HFE.

**Definition 4 (*Ren, Xu & Wang, 2018*).** *A NWHFS on a given universal set $\mathscr{Z} = \{z_1, z_2, \cdots, z_n\}$ is expressed mathematically by;*

$$NW = \{\langle z, \aleph(z), \Gamma(\aleph(z))\rangle | z \in \mathcal{Z}\}$$

*where $\aleph(z) = \{\tau_1, \tau_2, \cdots, \tau_{\#\aleph}\}$ is know as HFE, which can be symbolized as $\aleph$ and $\#\aleph$ denote the number of values in HFE. The function $\Gamma(\aleph(z))$ is designated as the NWE, enveloping the particulars of DM's preferences. The NWE indicated as $\Gamma(\aleph(z)) = \{\tilde{\tau}_1, \tilde{\tau}_2, \cdots, \tilde{\tau}_{\#\aleph}\}$ where $\tilde{\tau}_k = \{\alpha_k^L, \alpha_k^M, \alpha_k^U\}$ for all $k = \{1, 2, \cdots \#\aleph\}$, such that $\alpha_k^L, \alpha_k^M$ and $\alpha_k^U$ indicate the lower, middle and upper bounds of the preference degree respectively. The range $\tilde{\tau}_k = \{\alpha_k^L, \alpha_k^M, \alpha_k^U\}$ of HFE in NWE can be considered as:*

$$
\{\alpha_k^L, \alpha_k^M, \alpha_k^U\} = \left\{
\begin{array}{c}
\max\{(\tau_k - g(\tau_k)), 0\} \\
(-1 + 2rpd(\aleph(z_k)))g(\tau_k) + \tau_k, \\
\min\{(g(\tau_k) + \tau_k), 1\}
\end{array}
\right\}
\tag{2}
$$

*where the oscillatory function $g(\tau_k)$ is defined for regulating the membership degrees,*

$$g(\tau_k) = \sigma.e^{-\frac{(\tau_k - \bar{\aleph})^2}{2\sigma^2}}, \tag{3}$$

*it illustrates the fluctuating nature of preferences. The mean $\bar{\aleph}$ and the variance $\sigma$ of a HFE are expressed by*

$$\bar{\aleph} = \frac{1}{\#\aleph} \sum_{k=1}^{\#\aleph} \tau, \tag{4}$$

$$\sigma = \sqrt{\frac{1}{\#\aleph} \sum_{k=1}^{\#\aleph} (\tau_k - \overline{\aleph})^2}. \tag{5}$$

The real preference degree $(rpd)\aleph$ (*Ren, Xu & Wang, 2018*; *Yager, 1988*) of NWHFS is a key component which measures the pattern of DMs towards specific membership degrees. It is computed as follows:

$$rpd(\aleph) = \begin{cases} \sum_{i=1}^{\#\aleph} \overline{\tau}_i \left(\frac{\#\aleph - i}{\#\aleph - 1}\right) & \text{if } \overline{\aleph} < 0.5 \\ 1 - \sum_{i=1}^{\#\aleph} \overline{\tau}_i \left(\frac{\#\aleph - i}{\#\aleph - 1}\right) & \text{if } \overline{\aleph} > 0.5 \\ 0.5 & \text{if } \overline{\aleph} = 0.5 \end{cases} \tag{6}$$

where the values $\tau_i (i = 1, 2, \cdots \#\aleph)$ are normalized as, $\overline{\tau}_i = \tau_i / \left(\sum_{i=1}^{\#\aleph} \tau_i\right)$. Therefore, the pair $\langle \aleph(z), \Gamma(\aleph(z)) \rangle$ for $z \in \mathscr{Z}$ is known as the normal wiggly hesitant fuzzy element (NWHFE).

The concept of NWDHFS is investigated by *Narayanamoorthy et al. (2019)*, it is a contemporary mathematical instrument utilized to describe MGs and NMGs of the uncertain information inherent in the cognitive processes of DMs. It is expressed as follows;

**Definition 5 (*Narayanamoorthy et al., 2019*).** Let $\tilde{\mathscr{D}} = \{\langle z, \aleph(z), \Upsilon(z)\rangle \mid z \in \mathscr{Z}\}$ be a DHFS over $\mathscr{Z}$. A NWDHFS on a given universal set $\mathscr{Z}$ is given mathematically by;

$A = \{\langle z, \aleph(z), \Upsilon(z), \Gamma(\aleph(z)), \Psi(\Upsilon(z))\rangle | z \in \mathscr{Z}\}$

where $\aleph(z) = \bigcup_{\delta \in [0,1]} \{\delta\}$ and $\Upsilon(z) = \bigcup_{\gamma \in [0,1]} \{\gamma\}$ are called membership and non-memberhsip of $z \in \mathscr{Z}$. The functions $\Gamma(\aleph(z))$ and $\Psi(\Upsilon(z))$ are expressed as NWE for $\aleph(z)$ and $\Upsilon(z)$ respectively. Let $\#\aleph(z)$ denotes number of values in $\aleph(z)$. Then $\Gamma(\aleph(z)) = \{\tilde{\tau}_1, \tilde{\tau}_2, \cdots, \tilde{\tau}_{\#\aleph(z)}\}$, $\tilde{\tau}_k = \{\alpha_k^L, \alpha_k^M, \alpha_k^U\}$ $(k = 1, 2, \cdots, \#\aleph(z))$ such that

$$\{\alpha_k^L, \alpha_k^M, \alpha_k^U\} = \begin{cases} \max\{(\tau_k - g(\tau_k)), 0\} \\ \{-1 + 2rpd(\aleph(z_k))\}g(\tau_k) + \tau_k, \\ \min\{(g(\tau_k) + \tau_k), 1\} \end{cases}.$$

Furthermore, $\Psi(\Upsilon(z)) = \{\tilde{\delta}_1, \tilde{\delta}_2, \cdots, \tilde{\delta}_{\#\Upsilon(z)}\}$, $\tilde{\delta}_k = \{\beta_j^L, \beta_j^M, \beta_j^U\}$ $(j = 1, 2, \cdots \#\Upsilon(z))$ such that

$$\{\beta_j^L, \beta_j^M, \beta_j^U\} = \begin{cases} \max\{(\delta_j - g(\delta_j)), 0\} \\ \{-1 + 2rpd(\Upsilon(z_j))\}g(\delta_j) + \delta_j, \\ \min\{(g(\delta_j) + \delta_j), 1\} \end{cases}$$

where the functions $g(\tau_j), g(\delta_j)$ are calculated using the *Eqs. (3)–(5)* and the reference grades $rpd(\aleph), rpd(\Upsilon)$ are obtained using the *Eq. (6)*.

To quantitatively demonstrate the notion of NWDHFS, we use the following example.

**Table 1 Calculation of NWEs of DHFEs in $\tilde{D}$.**

| $\mathscr{Z}$ | Mean | | Standard deviation | | Wiggly function | | Preference degree | |
|---|---|---|---|---|---|---|---|---|
| | $\bar{\aleph}$ | $\bar{\Upsilon}$ | $\sigma_1$ | $\sigma_2$ | $g_1$ | $g_1$ | $rpd_1$ | $rpd_2$ |
| $z_1$ | 0.23 | 0.40 | 0.125 | 0.200 | 0.070, 0.120, 0.051 | 0.121, 0.121 | 0.286 | 0.250 |
| $z_2$ | 0.45 | 0.40 | 0.050 | 0.082 | 0.030, 0.030 | 0.039, 0.082, 0.039 | 0.444 | 0.420 |
| $z_3$ | 0.25 | 0.23 | 0.150 | 0.125 | 0.091, 0.091 | 0.070, 0.120, 0.051 | 0.364 | 0.286 |

**Example 2** *Let $\mathscr{Z} = \{z_1, z_2, z_3\}$ be a fix set and $\tilde{D}$ be a DHFS in it, then*

$$\tilde{D} = \left\{ \begin{array}{c} \langle z_1, (0.1, 0.2, 0.4), (0.2, 0.6)\rangle, \\ \langle z_2, (0.4, 0.5), (0.3, 0.4, 0.5)\rangle, \\ \langle z_3, (0.1, 0.4), (0.1, 0.2, 0.4)\rangle \end{array} \right\}.$$

*Utilizing Eqs. (3)–(6), we computed means, standard deviations, and preference degrees, which are presented in Table 1.*

Based on the calculation computed in Table 1 and using Definition (4), we formulated NWDHFS $A_{\tilde{D}}$ as follows;

$$A_{\tilde{D}} = \left\{ \begin{array}{c} \left( \begin{array}{c} z_1, (0.1, 0.2, 0.4), (0.2, 0.6), \\ \left\{ \begin{array}{c} (0.0296, 0.0698, 0.1704), (0.0797, 0.1484, 0.3203), \\ (0.349, 0.3781, 0.451) \end{array} \right\}, \\ \{(0.0787, 0.1394, 0.3213), (0.4787, 0.5394, 0.7213)\} \end{array} \right), \\ \left( \begin{array}{c} z_2, (0.4, 0.5), (0.3, 0.4, 0.5), \\ \{(0.3697, 0.3966, 0.4303), (0.4697, 0.4966, 0.5303)\}, \\ \left\{ \begin{array}{c} (0.2615, 0.2936, 0.3385), (0.3184, 0.3864, 0.4816), \\ (0.4615, 0.4936, 0.5385) \end{array} \right\} \end{array} \right), \\ \left( \begin{array}{c} z_3, (0.1, 0.4), (0.1, 0.2, 0.4), \\ \{(0.009, 0.1, 0.191), (0.309, 0.4, 0.491)\}, \\ \left\{ \begin{array}{c} (0.0296, 0.0698, 0.1704), (0.0797, 0.1484, 0.3203), \\ (0.349, 0.3781, 0.451) \end{array} \right\} \end{array} \right) \end{array} \right\}.$$

# CORRELATION COEFFICIENTS ON NORMAL WIGGLY DUAL HESITANT FUZZY SETS

The NWDHFS has enhanced preference ranges and resolved ambiguities among grades within DHFS. The establishment of CCs between two NWDHFSs is therefore required in order to establish a relationship between them. This section is dedicated to defining various characteristics of CCs, namely CCs and weighted CCs on NWDHFSs.

**Definition 6** *Let a reference set $\mathscr{Z} = \{z_1, z_2, \cdots, z_n\}$. Consider two NWDHFSs on $\mathscr{Z}$;*
*$A = \{\langle z, \aleph_A(z), \Upsilon_A(z), \Gamma_A(\aleph_A(z)), \Psi_A(\Upsilon_A(z))\rangle | z \in \mathscr{Z}\}$,*
*$B = \{\langle z, \aleph_B(z), \Upsilon_B(z), \Gamma_B(\aleph_B(z)), \Psi_B(\Upsilon_B(z))\rangle | z \in \mathscr{Z}\}$*

*where $\aleph_A(z_i) = \{\tau_{i1}, \tau_{i2}, \cdots, \tau_{i\#\aleph_A}\}$, $\Upsilon_A(z_i) = \{\delta_{i1}, \delta_{i2}, \cdots, \delta_{i\#\Upsilon_A}\}$,*
*$\aleph_B(z_i) = \{\eta_{i1}, \eta_{i2}, \cdots, \eta_{i\#\aleph_B}\}$, $\Upsilon_B(z_i) = \{\zeta_{i1}, \zeta_{i2}, \cdots, \zeta_{i\#\Upsilon_B}\}$, and $\Gamma_A(\aleph_A(z_i)) = \{\tilde{\tau}_{i1}, \tilde{\tau}_{i2}, \cdots, \tilde{\tau}_{i\#\aleph_A}\}$, $\Psi_A(\Upsilon_A(z_i)) = \{\tilde{\delta}_{i1}, \tilde{\delta}_{i2}, \cdots, \tilde{\delta}_{i\#\Upsilon_A}\}$ with $\tilde{\tau}_{ik} = \{\alpha_{ik}^L, \alpha_{ik}^M, \alpha_{ik}^U\}$*

$(k = 1, 2, \cdots, \#\aleph_A(z_i))$, $\tilde{\delta}_{ij} = \{\beta_{ij}^L, \beta_{ij}^M, \beta_{ij}^U\}$ $(j = 1, 2, \cdots, \#\Upsilon_A(z_i))$ and $\Gamma_B(\aleph_B(z_i)) = \{\breve{\eta}_1, \breve{\eta}_2, \cdots, \breve{\eta}_{i\#\aleph_B(z_i)}\}$, $\Psi_B(\Upsilon_B(z_i)) = \{\breve{\zeta}_1, \breve{\zeta}_2, \cdots, \breve{\zeta}_{\#\Upsilon_B(z_i)}\}$ with $\breve{\eta}_{ik'} = \{\mu_{ik'}^L, \mu_{ik'}^M, \mu_{ik'}^U\}$ $(k' = 1, 2, \cdots, \#\aleph_B(z_i))$, $\breve{\zeta}_{ij'} = \{v_{ij'}^L, v_{ij'}^M, v_{ij'}^U\}$ $(j' = 1, 2, \cdots, \#\Upsilon_B(z_i))$.

*Let $i\#\aleph_A, i\#\Upsilon_A, i\#\aleph_B$, and $i\#\Upsilon_B$ denote $\#\aleph_A(z_i), \#\Upsilon_A(z_i), \#\aleph_B(z_i)$ and $\#\Upsilon_B(z_i)$ for all $i \in (1, 2, \cdots, n)$. Then, the correlation between NWDHFSs A and B is expressed by;*

$$\mathfrak{C}(A, B) = \frac{1}{n}\sum_{i=1}^n \left[ \begin{array}{l} \left(\overline{\aleph}_A(z_i) - \overline{\Gamma}_A(\aleph(z_i))\right) \cdot \left(\overline{\aleph}_B(z_i) - \overline{\Gamma}_B(\aleph(z_i))\right) + \\ \left(\overline{\Upsilon}_A(z_i) - \overline{\Psi}_A(\Upsilon(z_i))\right) \cdot \left(\overline{\Upsilon}_B(z_i) - \overline{\Psi}_B(\Upsilon(z_i))\right) \end{array} \right] \tag{7}$$

*where the score functions are given in the following equations,*

$$\overline{\aleph}_A(z_i) = \frac{1}{i\#\aleph_A}\sum_{k=1}^{i\#\aleph_A} \tau_{Aik} \tag{8}$$

$$\overline{\aleph}_B(z_i) = \frac{1}{i\#\aleph_B}\sum_{k'=1}^{i\#\aleph_B} \eta_{Bik'} \tag{9}$$

$$\overline{\Upsilon}_A(z_i) = \frac{1}{i\#\Upsilon_A}\sum_{j=1}^{i\#\Upsilon_A} \delta_{Aij} \tag{10}$$

$$\overline{\Upsilon}_B(z_i) = \frac{1}{i\#\aleph_B}\sum_{j'=1}^{i\#\aleph_B} \zeta_{Bij'} \tag{11}$$

$$\overline{\Gamma}_A(\aleph_A(z_i)) = \frac{1}{i\#\aleph_A}\sum_{k=1}^{i\#\aleph_A} \overline{\tilde{\tau}_{Aik}} = \frac{1}{i\#\aleph_A}\sum_{k=1}^{i\#\aleph_A} \frac{\alpha_{Aik}^L + \alpha_{Aik}^M + \alpha_{Aik}^U}{3} \tag{12}$$

$$\overline{\Psi}_A(\Upsilon_A(z_i)) = \frac{1}{i\#\Upsilon_A}\sum_{j=1}^{i\#\Upsilon_A} \overline{\tilde{\delta}_{Aij}} = \frac{1}{i\#\Upsilon_A}\sum_{j=1}^{i\#\Upsilon_A} \frac{\beta_{Aij}^L + \beta_{Aij}^M + \beta_{Aij}^U}{3} \tag{13}$$

$$\overline{\Gamma}_B(\aleph_B(z_i)) = \frac{1}{i\#\aleph_B}\sum_{k'=1}^{i\#\aleph_B} \overline{\tilde{\eta}_{Bik'}} = \frac{1}{i\#\aleph_B}\sum_{k'=1}^{i\#\aleph_B} \frac{\mu_{Bik'}^L + \mu_{Bik'}^M + \mu_{Bik'}^U}{3} \tag{14}$$

$$\overline{\Psi}_B(\Upsilon_B(z_i)) = \frac{1}{i\#\Upsilon_B}\sum_{j'=1}^{i\#\Upsilon_B} \overline{\breve{\zeta}_{Bij'}} = \frac{1}{i\#\Upsilon_B}\sum_{j'=1}^{i\#\Upsilon_B} \frac{v_{Bij'}^L + v_{Bij'}^M + v_{Bij'}^U}{3}. \tag{15}$$

*Therefore, we can easily get correlation $\mathfrak{C}(A, A)$ as given below:*

$$\mathfrak{C}(A, A) = \frac{1}{n}\sum_{i=1}^n \left[ \left(\overline{\aleph}_A(z_i) - \overline{\Gamma}_A(\aleph(z_i))\right)^2 + \left(\overline{\Upsilon}_A(z_i) - \overline{\Psi}_A(\Upsilon(z_i))\right)^2 \right] \tag{16}$$

$$= \frac{1}{n}\sum_{i=1}^n \left[ \begin{array}{l} \left(\frac{1}{i\#\aleph_A}\sum_{k=1}^{i\#\aleph_A} \tau_{Aik} - \frac{1}{i\#\aleph_A}\sum_{k=1}^{i\#\aleph_A} \tilde{\tau}_{Aik}\right)^2 + \\ \left(\frac{1}{i\#\Upsilon_A}\sum_{j=1}^{i\#\Upsilon_A} \delta_{Aij} - \frac{1}{i\#\Upsilon_A}\sum_{j=1}^{i\#\Upsilon_A} \overline{\tilde{\delta}_{Aij}}\right)^2 \end{array} \right] \tag{17}$$

$$= \frac{1}{n}\sum_{i=1}^n \left[ \begin{array}{l} \left(\frac{1}{i\#\aleph_A}\sum_{k=1}^{i\#\aleph_A} \tau_{Aik} - \frac{1}{i\#\aleph_A}\sum_{k=1}^{i\#\aleph_A} \frac{\alpha_{Aik}^L + \alpha_{Aik}^M + \alpha_{Aik}^U}{3}\right)^2 + \\ \left(\frac{1}{i\#\Upsilon_A}\sum_{j=1}^{i\#\Upsilon_A} \delta_{Aij} - \frac{1}{i\#\Upsilon_A}\sum_{j=1}^{i\#\Upsilon_A} \frac{\beta_{Bij}^L + \beta_{Bij}^M + \beta_{Bij}^U}{3}\right)^2 \end{array} \right]. \tag{18}$$

Subsequently, utilizing the concepts of $\mathfrak{C}(A, B)$ and $\mathfrak{C}(A, A)$, we defined CC on NWDHFSs in the following approach.

**Definition 7** *Let a reference set $\mathscr{Z} = \{z_1, z_2, \cdots, z_n\}$. Consider two NWDHFSs on $\mathscr{Z}$;*

$$A = \{\langle z, \aleph_A(z), \Upsilon_A(z), \Gamma_A(\aleph_A(z)), \Psi_A(\Upsilon_A(z)) \rangle | z \in \mathscr{Z}\},$$
$$B = \{\langle z, \aleph_B(z), \Upsilon_B(z), \Gamma_B(\aleph_B(z)), \Psi_B(\Upsilon_B(z)) \rangle | z \in \mathscr{Z}\}$$

*where* $\aleph_A(z_i) = \{\tau_{i1}, \tau_{i2}, \cdots, \tau_{i\#\aleph_A}\}$, $\Upsilon_A(z_i) = \{\delta_{i1}, \delta_{i2}, \cdots, \delta_{i\#\Upsilon_A}\}$,
$\aleph_B(z_i) = \{\eta_{i1}, \eta_{i2}, \cdots, \eta_{i\#\aleph_B}\}$, $\Upsilon_B(z_i) = \{\zeta_{i1}, \zeta_{i2}, \cdots, \zeta_{i\#\Upsilon_B}\}$, *and*
$\Gamma_A(\aleph_A(z_i)) = \{\tilde{\tau}_{i1}, \tilde{\tau}_{i2}, \cdots, \tilde{\tau}_{i\#\aleph_A}\}$, $\Psi_A(\Upsilon_A(z_i)) = \{\tilde{\delta}_{i1}, \tilde{\delta}_{i2}, \cdots, \tilde{\delta}_{i\#\Upsilon_A}\}$ *with*
$\tilde{\tau}_k = \{\alpha_k^L, \alpha_k^M, \alpha_k^U\}$ $(k = 1, 2, \cdots \#\aleph_A(z_i))$, $\tilde{\delta}_j = \{\delta_j^L, \delta_j^M, \delta_j^U\}$ $(j = 1, 2, \cdots \#\Upsilon_A(z_i))$ *and*
$\Gamma_B(\aleph_B(z_i)) = \{\breve{\eta}_1, \breve{\eta}_2, \cdots, \breve{\eta}_{\#\aleph_B(z_i)}\}$, $\Psi_B(\Upsilon_B(z_i)) = \{\breve{\zeta}_1, \breve{\zeta}_2, \cdots, \breve{\zeta}_{\#\Upsilon_B(z_i)}\}$ *with*
$\breve{\eta}_{k'} = \{\eta_{k'}^L, \eta_{k'}^M, \eta_{k'}^U\}$ $(k' = 1, 2, \cdots \#\aleph_B(z_i))$, $\breve{\zeta}_{j'} = \{\zeta_{j'}^L, \zeta_{j'}^M, \zeta_{j'}^U\}$ $(j' = 1, 2, \cdots \#\Upsilon_B(z_i))$.
*The correlation coefficient between two NWDHFSs A and B is given by;*

$$\Delta_{NWDH_1}(A, B) = \frac{\mathfrak{C}(A, B)}{\sqrt{\mathfrak{C}(A, A)} \cdot \sqrt{\mathfrak{C}(B, B)}} \tag{19}$$

$$= \frac{\frac{1}{n}\sum_{i=1}^{n}\left[\begin{array}{l}(\bar{\aleph}_A(z_i) - \bar{\Gamma}_A(\aleph(z_i))) \cdot (\bar{\aleph}_B(z_i) - \bar{\Gamma}_B(\aleph(z_i))) \\ + (\bar{\Upsilon}_A(z_i) - \bar{\Psi}_A(\Upsilon(z_i))) \cdot (\bar{\Upsilon}_B(z_i) - \bar{\Psi}_B(\Upsilon(z_i)))\end{array}\right]}{\sqrt{\frac{1}{n}\sum_{i=1}^{n}\left[\begin{array}{l}(\bar{\aleph}_A(z_i) - \bar{\Gamma}_A(\aleph(z_i)))^2 \\ + (\bar{\Upsilon}_A(z_i) - \bar{\Psi}_A(\Upsilon(z_i)))^2\end{array}\right]} \cdot \sqrt{\frac{1}{n}\sum_{i=1}^{n}\left[\begin{array}{l}(\bar{\aleph}_B(z_i) - \bar{\Gamma}_B(\aleph(z_i)))^2 \\ + (\bar{\Upsilon}_B(z_i) - \bar{\Psi}_B(\Upsilon(z_i)))^2\end{array}\right]}}. \tag{20}$$

**Theorem 1** *The formula of CC in Eq. (19) between two NWDHFSs verifies the given properties:*

(I) $\Delta_{NWDH_1}(A, B) = \Delta_{NWDH_1}(B, A)$
(II) $\Delta_{NWDH_1}(A, A) = 1$
(III) $|\Delta_{NWDH_1}(A, B)| \leq 1$

**Proof:**

(I) Let us consider the CC between two NWDHFSs $A$ and $B$. Then,

$$\Delta_{NWDH_1}(A, B) = \frac{\frac{1}{n}\sum_{i=1}^{n}\left[\begin{array}{l}(\bar{\aleph}_A(z_i) - \bar{\Gamma}_A(\aleph(z_i))) \cdot (\bar{\aleph}_B(z_i) - \bar{\Gamma}_B(\aleph(z_i))) \\ + (\bar{\Upsilon}_A(z_i) - \bar{\Psi}_A(\Upsilon(z_i))) \cdot (\bar{\Upsilon}_B(z_i) - \bar{\Psi}_B(\Upsilon(z_i)))\end{array}\right]}{\sqrt{\frac{1}{n}\sum_{i=1}^{n}\left[\begin{array}{l}(\bar{\aleph}_A(z_i) - \bar{\Gamma}_A(\aleph(z_i)))^2 \\ + (\bar{\Upsilon}_A(z_i) - \bar{\Psi}_A(\Upsilon(z_i)))^2\end{array}\right]} \cdot \sqrt{\frac{1}{n}\sum_{i=1}^{n}\left[\begin{array}{l}(\bar{\aleph}_B(z_i) - \bar{\Gamma}_B(\aleph(z_i)))^2 \\ + (\bar{\Upsilon}_B(z_i) - \bar{\Psi}_B(\Upsilon(z_i)))^2\end{array}\right]}}$$

$$= \frac{\frac{1}{n}\sum_{i=1}^{n}\left[\begin{array}{l}(\bar{\aleph}_B(z_i) - \bar{\Gamma}_B(\aleph(z_i))) \cdot (\bar{\aleph}_A(z_i) - \bar{\Gamma}_A(\aleph(z_i))) \\ + (\bar{\Upsilon}_B(z_i) - \bar{\Psi}_B(\Upsilon(z_i))) \cdot (\bar{\Upsilon}_A(z_i) - \bar{\Psi}_A(\Upsilon(z_i)))\end{array}\right]}{\sqrt{\frac{1}{n}\sum_{i=1}^{n}\left[\begin{array}{l}(\bar{\aleph}_B(z_i) - \bar{\Gamma}_B(\aleph(z_i)))^2 \\ + (\bar{\Upsilon}_B(z_i) - \bar{\Psi}_B(\Upsilon(z_i)))^2\end{array}\right]} \cdot \sqrt{\frac{1}{n}\sum_{i=1}^{n}\left[\begin{array}{l}(\bar{\aleph}_A(z_i) - \bar{\Gamma}_A(\aleph(z_i)))^2 \\ + (\bar{\Upsilon}_A(z_i) - \bar{\Psi}_A(\Upsilon(z_i)))^2\end{array}\right]}}$$

$$= \Delta_{NWDH_1}(B, A)$$

(II) The second condition is demonstrated by the following.

$$\Delta_{NWDH_1}(A,A) = \frac{\frac{1}{n}\sum_{i=1}^{n}\left[\begin{array}{c}\left(\bar{\aleph}_A(z_i) - \bar{\Gamma}_A(\aleph(z_i))\right) \cdot \left(\bar{\aleph}_A(z_i) - \bar{\Gamma}_A(\aleph(z_i))\right) \\ + \left(\bar{\Upsilon}_A(z_i) - \bar{\Psi}_A(\Upsilon(z_i))\right) \cdot \left(\bar{\Upsilon}_A(z_i) - \bar{\Psi}_A(\Upsilon(z_i))\right)\end{array}\right]}{\sqrt{\frac{1}{n}\sum_{i=1}^{n}\left[\begin{array}{c}\left(\bar{\aleph}_A(z_i) - \bar{\Gamma}_A(\aleph(z_i))\right)^2 \\ + \left(\bar{\Upsilon}_A(z_i) - \bar{\Psi}_A(\Upsilon(z_i))\right)^2\end{array}\right]} \cdot \sqrt{\frac{1}{n}\sum_{i=1}^{n}\left[\begin{array}{c}\left(\bar{\aleph}_A(z_i) - \bar{\Gamma}_A(\aleph(z_i))\right)^2 \\ + \left(\bar{\Upsilon}_A(z_i) - \bar{\Psi}_A(\Upsilon(z_i))\right)^2\end{array}\right]}}$$

$$= \frac{\frac{1}{n}\sum_{i=1}^{n}\left[\begin{array}{c}\left(\bar{\aleph}_A(z_i) - \bar{\Gamma}_A(\aleph(z_i))\right)^2 \\ + \left(\bar{\Upsilon}_A(z_i) - \bar{\Psi}_A(\Upsilon(z_i))\right)^2\end{array}\right]}{\frac{1}{n}\sum_{i=1}^{n}\left[\begin{array}{c}\left(\bar{\aleph}_A(z_i) - \bar{\Gamma}_A(\aleph(z_i))\right)^2 \\ + \left(\bar{\Upsilon}_A(z_i) - \bar{\Psi}_A(\Upsilon(z_i))\right)^2\end{array}\right]}$$

$$= 1.$$

(III) Now we finally prove the third property of the theorem;

$$|\mathfrak{C}(A,B)|$$

$$= \left|\frac{1}{n}\sum_{i=1}^{n}\left[\begin{array}{c}\left(\bar{\aleph}_A(z_i) - \bar{\Gamma}_A(\aleph(z_i))\right) \cdot \left(\bar{\aleph}_B(z_i) - \bar{\Gamma}_B(\aleph(z_i))\right) \\ + \left(\bar{\Upsilon}_A(z_i) - \bar{\Psi}_A(\Upsilon(z_i))\right) \cdot \left(\bar{\Upsilon}_B(z_i) - \bar{\Psi}_B(\Upsilon(z_i))\right)\end{array}\right]\right|$$

$$\leq \frac{1}{n}\sum_{i=1}^{n}\left|\left(\bar{\aleph}_A(z_i) - \bar{\Gamma}_A(\aleph(z_i))\right) \cdot \left(\bar{\aleph}_B(z_i) - \bar{\Gamma}_B(\aleph(z_i))\right)\right|$$

$$+ \frac{1}{n}\sum_{i=1}^{n}\left|\left(\bar{\Upsilon}_A(z_i) - \bar{\Psi}_A(\Upsilon(z_i))\right) \cdot \left(\bar{\Upsilon}_B(z_i) - \bar{\Psi}_B(\Upsilon(z_i))\right)\right|$$

$$\leq \frac{1}{n}\sum_{i=1}^{n}\left|\left(\bar{\aleph}_A(z_i) - \bar{\Gamma}_A(\aleph(z_i))\right)\right| \cdot \left|\left(\bar{\aleph}_B(z_i) - \bar{\Gamma}_B(\aleph(z_i))\right)\right|$$

$$+ \frac{1}{n}\sum_{i=1}^{n}\left|\left(\bar{\Upsilon}_A(z_i) - \bar{\Psi}_A(\Upsilon(z_i))\right)\right| \cdot \left|\left(\bar{\Upsilon}_B(z_i) - \bar{\Psi}_B(\Upsilon(z_i))\right)\right|$$

$$\leq \left[\frac{1}{n}\sum_{i=1}^{n}\left|\left(\bar{\aleph}_A(z_i) - \bar{\Gamma}_A(\aleph(z_i))\right)\right|^2 \cdot \frac{1}{n}\sum_{i=1}^{n}\left|\left(\bar{\aleph}_B(z_i) - \bar{\Gamma}_B(\aleph(z_i))\right)\right|^2\right]^{\frac{1}{2}}$$

$$+ \left[\frac{1}{n}\sum_{i=1}^{n}\left|\left(\bar{\Upsilon}_A(z_i) - \bar{\Psi}_A(\Upsilon(z_i))\right)\right|^2 \cdot \frac{1}{n}\sum_{i=1}^{n}\left|\left(\bar{\Upsilon}_B(z_i) - \bar{\Psi}_B(\Upsilon(z_i))\right)\right|^2\right]^{\frac{1}{2}}$$

$$= \left[\begin{array}{c}\left[\frac{1}{n}\sum_{i=1}^{n}\left|\left(\bar{\aleph}_A(z_i) - \bar{\Gamma}_A(\aleph(z_i))\right)\right|^2\right]^{\frac{1}{2}} \cdot \left[\frac{1}{n}\sum_{i=1}^{n}\left|\left(\bar{\aleph}_B(z_i) - \bar{\Gamma}_B(\aleph(z_i))\right)\right|^2\right]^{\frac{1}{2}} \\ \left[\frac{1}{n}\sum_{i=1}^{n}\left|\left(\bar{\Upsilon}_A(z_i) - \bar{\Psi}_A(\Upsilon(z_i))\right)\right|^2\right]^{\frac{1}{2}} \cdot \left[\frac{1}{n}\sum_{i=1}^{n}\left|\left(\bar{\Upsilon}_B(z_i) - \bar{\Psi}_B(\Upsilon(z_i))\right)\right|^2\right]^{\frac{1}{2}}\end{array}\right]^{\frac{1}{2}}$$

By Cauchy-Schwarz inequality see footnote[1]

$$\leq \left[\begin{array}{c}\left[\left[\frac{1}{n}\sum_{i=1}^{n}\left|\left(\bar{\aleph}_A(z_i) - \bar{\Gamma}_A(\aleph(z_i))\right)\right|^2\right] + \left[\frac{1}{n}\sum_{i=1}^{n}\left|\left(\bar{\Upsilon}_A(z_i) - \bar{\Psi}_A(\Upsilon(z_i))\right)\right|^2\right]\right] \cdot \\ \left[\left[\frac{1}{n}\sum_{i=1}^{n}\left|\left(\bar{\aleph}_B(z_i) - \bar{\Gamma}_B(\aleph(z_i))\right)\right|^2\right] + \left[\frac{1}{n}\sum_{i=1}^{n}\left|\left(\bar{\Upsilon}_B(z_i) - \bar{\Psi}_B(\Upsilon(z_i))\right)\right|^2\right]\right]\end{array}\right]^{\frac{1}{2}}$$

$$= \left[\mathfrak{C}(A,A) \cdot \mathfrak{C}(B,B)\right]^{\frac{1}{2}}$$

$$\Delta_{NWDH_1}(A,B) = \frac{\mathfrak{C}(A,B)}{\sqrt{\mathfrak{C}(A,A)} \cdot \sqrt{\mathfrak{C}(B,B)}} \leq 1.$$

[1] For real numbers $p_{i'}, q_{i'}$ ($i' = \{1, 2, \cdots, n'\}$). Cauchy-Schwarz inequality is given by $(p_1q_1 + p_2q_2 + \cdots + p_{n'}q_{n'})^2 \leq (p_1^2 + p_2^2 + \cdots + p_{n'}^2) \cdot (q_1^2 + q_2^2 + \cdots + q_{n'}^2)$.

According to Eq. (19), we determined the CC of NWDHFSs by taking the product of $\mathfrak{C}(A, A)$ and $\mathfrak{C}(B, B)$ in the denominator. We will only consider the maximum value from $\mathfrak{C}^2(A, A)$ and $\mathfrak{C}^2(B, B)$ in the subsequent computation of the CC.

**Definition 8** *Let a reference set* $\mathscr{Z} = \{z_1, z_2, \cdots, z_n\}$. *Consider two NWDHFSs on* $\mathscr{Z}$;

$A = \{\langle z, \aleph_A(z), \Upsilon_A(z), \Gamma_A(\aleph_A(z)), \Psi_A(\Upsilon_A(z))\rangle | z \in \mathscr{Z}\}$,
$B = \{\langle z, \aleph_B(z), \Upsilon_B(z), \Gamma_B(\aleph_B(z)), \Psi_B(\Upsilon_B(z))\rangle | z \in \mathscr{Z}\}$

*where* $\aleph_A(z_i) = \{\tau_{i1}, \tau_{i2}, \cdots, \tau_{i\#\aleph_A}\}$, $\Upsilon_A(z_i) = \{\delta_{i1}, \delta_{i2}, \cdots, \delta_{i\#\Upsilon_A}\}$,
$\aleph_B(z_i) = \{\eta_{i1}, \eta_{i2}, \cdots, \eta_{i\#\aleph_B}\}$, $\Upsilon_B(z_i) = \{\zeta_{i1}, \zeta_{i2}, \cdots, \zeta_{i\#\Upsilon_B}\}$, *and*
$\Gamma_A(\aleph_A(z_i)) = \{\tilde{\tau}_{i1}, \tilde{\tau}_{i2}, \cdots, \tilde{\tau}_{i\#\aleph_A}\}$, $\Psi_A(\Upsilon_A(z_i)) = \{\tilde{\delta}_{i1}, \tilde{\delta}_{i2}, \cdots, \tilde{\delta}_{i\#\Upsilon_A}\}$ *with*
$\tilde{\tau}_k = \{\alpha_k^L, \alpha_k^M, \alpha_k^U\}$ $(k = 1, 2, \cdots \#\aleph_A(z_i))$, $\tilde{\delta}_j = \{\delta_j^L, \delta_j^M, \delta_j^U\}$ $(j = 1, 2, \cdots \#\Upsilon_A(z_i))$ *and*
$\Gamma_B(\aleph_B(z_i)) = \{\breve{\eta}_1, \breve{\eta}_2, \cdots, \breve{\eta}_{\#\aleph_B(z_i)}\}$, $\Psi_B(\Upsilon_B(z_i)) = \{\breve{\zeta}_1, \breve{\zeta}_2, \cdots, \breve{\zeta}_{\#\Upsilon_B(z_i)}\}$ *with*
$\breve{\eta}_{k'} = \{\eta_{k'}^L, \eta_{k'}^M, \eta_{k'}^U\}$ $(k' = 1, 2, \cdots \#\aleph_B(z_i))$, $\breve{\zeta}_{j'} = \{\zeta_{j'}^L, \zeta_{j'}^M, \zeta_{j'}^U\}$ $(j' = 1, 2, \cdots \#\Upsilon_B(z_i))$.

*The CCs between two NWDHFSs A and B is investigated by;*

$$\Delta_{NWDH_2}(A, B) \tag{21}$$

$$= \frac{\mathfrak{C}(A, B)}{\max\{\mathfrak{C}(A, A), \mathfrak{C}(B, B)\}} \tag{22}$$

$$= \frac{\frac{1}{n}\sum_{i=1}^{n}\left[\begin{array}{c}(\overline{\aleph}_A(z_i) - \overline{\Gamma}_A(\aleph(z_i))) \cdot (\overline{\aleph}_B(z_i) - \overline{\Gamma}_B(\aleph(z_i))) \\ +(\overline{\Upsilon}_A(z_i) - \overline{\Psi}_A(\Upsilon(z_i))) \cdot (\overline{\Upsilon}_B(z_i) - \overline{\Psi}_B(\Upsilon(z_i)))\end{array}\right]}{\max\left\{\frac{1}{n}\sum_{i=1}^{n}\left[\begin{array}{c}(\overline{\aleph}_A(z_i) - \overline{\Gamma}_A(\aleph(z_i)))^2 \\ +(\overline{\Upsilon}_A(z_i) - \overline{\Psi}_A(\Upsilon(z_i)))^2\end{array}\right], \frac{1}{n}\sum_{i=1}^{n}\left[\begin{array}{c}(\overline{\aleph}_B(z_i) - \overline{\Gamma}_B(\aleph(z_i)))^2 \\ +(\overline{\Upsilon}_B(z_i) - \overline{\Psi}_B(\Upsilon(z_i)))^2\end{array}\right]\right\}}. \tag{23}$$

**Theorem 2** *The CCs on NWDHFSs expressed in Definitions (7) and (8) are compared as follows;* $|\Delta_{NWDH_2}(A, B)| \leq |\Delta_{NWDH_1}(A, B)|$.

**Proof:** We know that for real numbers $p, q \in [0, 1]$ hold inequality $p.q \leq \max\{p, q\}$. Therefore

$$\sqrt{\mathfrak{C}(A, A)} \cdot \sqrt{\mathfrak{C}(B, B)} \leq \sqrt{\max\{\mathfrak{C}^2(A, A), \mathfrak{C}^2(B, B)\}}$$

$$\Rightarrow \quad \sqrt{\mathfrak{C}(A, A)} \cdot \sqrt{\mathfrak{C}(B, B)} \leq \max\{\mathfrak{C}(A, A), \mathfrak{C}(B, B)\}$$

$$\Rightarrow \quad \frac{1}{\sqrt{\mathfrak{C}(A, A)} \cdot \sqrt{\mathfrak{C}(B, B)}} \geq \frac{1}{\max\{\mathfrak{C}(A, A), \mathfrak{C}(B, B)\}}$$

$$\Rightarrow \quad \frac{\mathfrak{C}(A, B)}{\sqrt{\mathfrak{C}(A, A)} \cdot \sqrt{\mathfrak{C}(B, B)}} \geq \frac{\mathfrak{C}(A, B)}{\max\{\mathfrak{C}(A, A), \mathfrak{C}(B, B)\}}$$

$$\Rightarrow \quad |\Delta_{NWDH_2}(A, B)| \leq |\Delta_{NWDH_1}(A, B)|.$$

**Theorem 3** *The CC in the Definition (8) satisfies the following conditions:*

(I) $\Delta_{NWDH_2}(A, B) = \Delta_{NWDH2}(B, A)$,

(II) $\Delta_{NWDH_2}(A, A) = 1,$

(III) $|\Delta_{NWDH_2}(A, B)| \leq 1.$

**Proof:** Proof of property (I) and (II) is similar to the proof of axiom (I) and (II) of the Theorem 1. It is required to prove property (III). In the Theorem 2 we proved that

$$|\Delta_{NWDH_2}(A, B)| \leq |\Delta_{NWDH_1}(A, B)|. \tag{24}$$

Furthermore in Theorem 1, it is given that $|\Delta_{NWDH_1}(A, B)| \leq 1$. Then by the Eq. (24), we have

$$|\Delta_{NWDH_2}(A, B)| \leq |\Delta_{NWDH_1}(A, B)| \leq 1.$$

Hence $|\Delta_{NWDH_2}(A, B)| \leq 1$.

In real world, $z_i \in \mathscr{Z}(i = 1, 2, \cdots, n)$ may have different level of importance under different circumstances. Keeping this fact in mind, DMs will give different weights to the $z_i \in \mathscr{Z}$. On the basis of this assumption of the DMs, we present the weighted CC between two NWDHFSs in next result.

**Definition 9** *Consider a reference set $\mathscr{Z} = \{z_1, z_2, \cdots, z_n\}$. Let $\varsigma = \{\varsigma_1, \varsigma_2, \cdots, \varsigma_n\}$ be a vector of weights for $z_i \in \mathscr{Z}$, with $\varsigma_i > 0$ and $\sum\limits_{i=1}^{n} \varsigma_i = 1$. Let two NWDHFSs*

$$A = \{\langle z, \aleph_A(z), \Upsilon_A(z), \Gamma_A(\aleph_A(z)), \Psi_A(\Upsilon_A(z))\rangle | z \in \mathscr{Z}\}$$
$$B = \{\langle z, \aleph_B(z), \Upsilon_B(z), \Gamma_B(\aleph_B(z)), \Psi_B(\Upsilon_B(z))\rangle | z \in \mathscr{Z}\} \ on \ \mathscr{Z},$$

*The weighted CC between them A and B is investigated as follow.*

$$\Delta_{NWDH_{\varsigma 1}}(A, B) \tag{25}$$

$$= \frac{\mathfrak{C}_\varsigma(A, B)}{\sqrt{\mathfrak{C}_\varsigma(A, A)}, \sqrt{\mathfrak{C}_\varsigma(B, B)}} \tag{26}$$

$$= \frac{\frac{1}{n}\sum\limits_{i=1}^{n} \varsigma_i \left[ \begin{array}{l} \left(\bar{\aleph}_A(z_i) - \bar{\Gamma}_A(\aleph(z_i))\right) \cdot \left(\bar{\aleph}_B(z_i) - \bar{\Gamma}_B(\aleph(z_i))\right) \\ + \left(\bar{\Upsilon}_A(z_i) - \bar{\Psi}_A(\Upsilon(z_i))\right) \cdot \left(\bar{\Upsilon}_B(z_i) - \bar{\Psi}_B(\Upsilon(z_i))\right) \end{array} \right]}{\sqrt{\frac{1}{n}\sum\limits_{i=1}^{n} \varsigma_i \left[ \begin{array}{l} \left(\bar{\aleph}_A(z_i) - \bar{\Gamma}_A(\aleph(z_i))\right)^2 \\ + \left(\bar{\Upsilon}_A(z_i) - \bar{\Psi}_A(\Upsilon(z_i))\right)^2 \end{array} \right]} \cdot \sqrt{\frac{1}{n}\sum\limits_{i=1}^{n} \varsigma_i \left[ \begin{array}{l} \left(\bar{\aleph}_B(z_i) - \bar{\Gamma}_B(\aleph(z_i))\right)^2 \\ + \left(\bar{\Upsilon}_B(z_i) - \bar{\Psi}_B(\Upsilon(z_i))\right)^2 \end{array} \right]}}. \tag{27}$$

*Another, weighted CC is expressed as in follows.*

$$\Delta_{NWDH_{\varsigma 2}}(A, B) \tag{28}$$

$$= \frac{\mathfrak{C}_\varsigma(A, B)}{\max\{\mathfrak{C}_\varsigma(A, A), \mathfrak{C}_\varsigma(B, B)\}} \tag{29}$$

$$= \frac{\frac{1}{n}\sum\limits_{i=1}^{n} \varsigma_i \left[ \begin{array}{l} \left(\bar{\aleph}_A(z_i) - \bar{\Gamma}_A(\aleph(z_i))\right) \cdot \left(\bar{\aleph}_B(z_i) - \bar{\Gamma}_B(\aleph(z_i))\right) \\ + \left(\bar{\Upsilon}_A(z_i) - \bar{\Psi}_A(\Upsilon(z_i))\right) \cdot \left(\bar{\Upsilon}_B(z_i) - \bar{\Psi}_B(\Upsilon(z_i))\right) \end{array} \right]}{\max \left\{ \begin{array}{l} \frac{1}{n}\sum\limits_{i=1}^{n} \varsigma_i \left[ \begin{array}{l} \left(\bar{\aleph}_A(z_i) - \bar{\Gamma}_A(\aleph(z_i))\right)^2 \\ + \left(\bar{\Upsilon}_A(z_i) - \bar{\Psi}_A(\Upsilon(z_i))\right)^2 \end{array} \right], \\ \frac{1}{n}\sum\limits_{i=1}^{n} \varsigma_i \left[ \begin{array}{l} \left(\bar{\aleph}_B(z_i) - \bar{\Gamma}_B(\aleph(z_i))\right)^2 \\ + \left(\bar{\Upsilon}_B(z_i) - \bar{\Psi}_B(\Upsilon(z_i))\right)^2 \end{array} \right] \end{array} \right\}}. \tag{30}$$

---

| Algorithm 1 | Method on CCs of NWDHFSs $\Delta_{NWDH_1}$ or $\Delta_{NWDH_2}$. |
|---|---|

1: Consider a group of possibilities/alternatives denoted by $\mathscr{A} = \{\mathscr{A}_1, \mathscr{A}_2, \cdots, \mathscr{A}_s\}$ and a group of attributes denoted by $\mathscr{C} = \{\mathscr{C}_1, \mathscr{C}_2, \cdots, \mathscr{C}_n\}$. A committee of specialists presented evaluations in the form of DHFE on $\mathscr{A}$, encompassing attributes. A decision matrix is derived and designated as $[DHF]_{n \times s}$.

2: Obtain a reference set $\mathscr{Y}^*$ which contained realistic assessment data in the form of DHFEs, it is widely recognized among experts.

3: Normalize decision matrix $[DHF]_{n \times s}$ utilizing the following equation,

$$\langle \aleph_{\tilde{\mathscr{D}}}(\mathscr{C}_i), \Upsilon_{\tilde{\mathscr{D}}}(\mathscr{C}_i) \rangle = \begin{cases} \langle \Re_{\tilde{\mathscr{D}}}(\mathscr{C}_i), \Im_{\tilde{\mathscr{D}}}(\mathscr{C}_i) \rangle & \text{if } \mathscr{C}_i \text{ is beneficial type} \\ \langle \Im_{\tilde{\mathscr{D}}}(\mathscr{C}_i), \Re_{\tilde{\mathscr{D}}}(\mathscr{C}_i) \rangle & \text{if } \mathscr{C}_i \text{ is cost type} \end{cases} \qquad (31)$$

4: Utilize the Definition of NWDHFS to transform $[DHFE]_{n \times s}$ into $[NWDHFE]_{n \times s}$. Similarly, transfer $\mathscr{Y}^*$ to NWDHFS($\mathscr{Y}^*$).

5: Utilizing Eqs. (8)–(15) calculate mean values of data in $[NWDHF]_{n \times s}$ and NWDHFS($\mathscr{Y}^*$) and write them in Table $t_1$ and Table $t_2$ respectively.

6: Compute CCs NWDHFSs $\Delta_{NWDH_1}$ or $\Delta_{NWDH_2}$ on Table $t_1$ and Table $t_2$.

7: Order the alternatives based on the results from CC on NWDHFSs $\Delta_{NWDH_1}$ or $\Delta_{NWDH_2}$.

8: The optimal selection from a set of alternatives is determined by the highest value of CC on NWDHFSs.

## MCDM APPROACH BASED ON CORRELATION COEFFICIENTS BETWEEN NWDHFSS

An appropriate MCDM method is essential for managing uncertainty in all aspects under challenging real-world conditions where making decisions based on specified criteria is problematic (*Dagıstanlı, 2023*; *Kumar & Pamucar, 2025*; *Radenovic et al., 2023*). An effective technique is essential for the MCDM method concerning NWDHFSs, as NWDHFSs include dualism and hesitancy in complex decision-making processes.

This section will demonstrate the MCDM procedure and provide an adequate illustration utilizing enhanced correlation coefficients on NWDHFSs.

The method that we developed based on NWDHFSs is comprised of the steps in the Algorithms 1 and 2. The decision-making process is illustrated in Fig. 1 by a flow diagram.

## APPLICATION OF THE METHOD FOR THE SELECTION OF REAL ESTATE AGENT

Real estate is a type of real property that consists of land and anything permanently attached to it, such as buildings, roads, fixtures, and utility systems. It can also include natural resources like water, minerals, plants, and animals. Real estate can be used for residential, commercial, or industrial purposes. When making investment decisions, real estate consultants are essential since they provide access to opportunities, risk assessment, and insightful market information. In the complicated world of real estate, their knowledge, insight, and connections give investors a competitive edge. A real estate agent is a certified expert who connects buyers and sellers, organizes real estate transactions, and negotiates on their behalf. The size and quantity of transactions that real estate brokers conclude determine how much money they earn because they are typically compensated with a commission, which is a portion of the sale price of the real estate. The experience of purchasing or selling a property can be greatly impacted by the choice of real estate agent. When selecting a real estate agent, keep the following important key factors in mind.

**Algorithm 2** Method on weighted CCs of NWDHFSs $\Delta_{NWDH_{c1}}$ or $\Delta_{NWDH_{c2}}$.

1:   This algorithm addresses steps 1 through 5 of the aforementioned Algorithm 1.

2:   Consider weights $\varsigma_1, \varsigma_2, \cdots, \varsigma_n$ for $\mathscr{C}_i(i = 1, 2, \cdots, n)$ respectively.

3:   Compute weighted CCs on NWDHFSs $\Delta_{NWDH_{c1}}$ or $\Delta_{NWDH_{c2}}$ on Table $t_1$ and Table $t_2$.

4:   Order the alternatives based on the results from weighted CCs on NWDHFSs $\Delta_{NWDH_{c1}}$ or $\Delta_{NWDH_{c2}}$.

5:   The optimal selection from a set of alternatives is determined by the maximum value of weighted NWDHFSs.

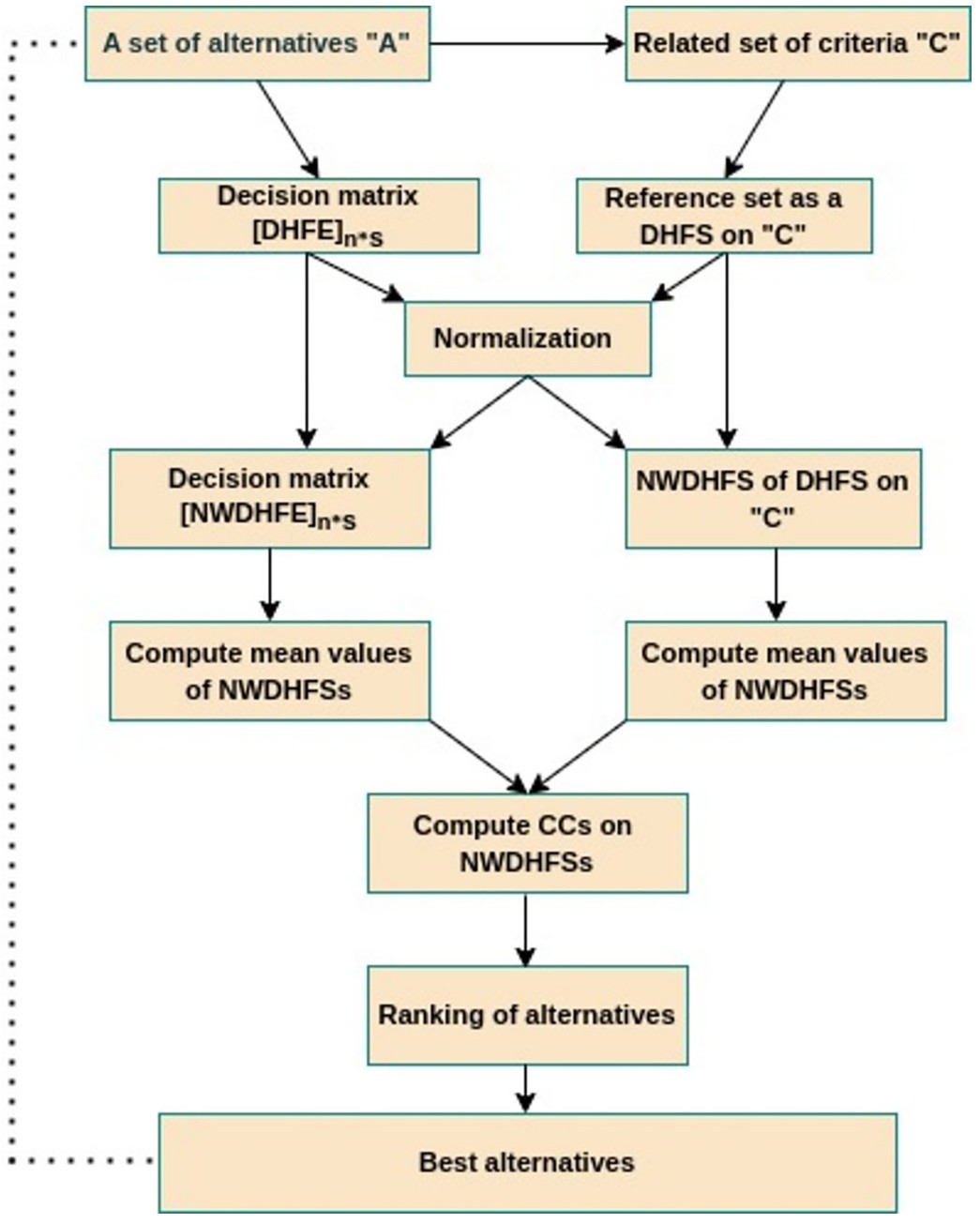

**Figure 1** MCDM approach based on correlation coefficients between NWDHFSs.

(a) Tech-savy ($\mathscr{C}_1$): It is quite advantageous in modern times to have an agent who uses digital tools for marketing or property discovery. You want to find a person who uses virtual tours, social networking, and internet advertising.

(b) Network & resources ($\mathscr{C}_2$): Is the agent in contact with reputable contractors, inspectors, or financial brokers? One advantage may be their network.

(c) Negotiation skills ($\mathscr{C}_3$): A good negotiator can assist you accomplish the best price when selling or save money while buying.

(d) Experience & Expertise ($\mathscr{C}_4$): In your local market, look for an agent with a track record of success. A real estate agent that knows your region well will have important knowledge about neighborhoods, prices, and market trends because real estate is frequently hyper-local.

These elements are accepted as selection criteria for real estate agents. In order to engage in the commercial and industrial sectors, a real estate company needs a competent real estate agent because these kinds of investments demand a number of important characteristics, including, research the market, financing, negotiation, site visits, post-purchase considerations and *etc.*

**Step-1.** There are four real state agents, represented in the set $\mathscr{A} = \{\mathscr{A}_{i'} \mid i' = 1, 2, \cdots, 4\}$, from them one will be chosen for investments in commercial and industrial sectors. Regarding the specified criteria, three real estate specialists offered their assessments of $\mathscr{A}_{i'}(i' = 1, 2, \cdots, 4)$; these assessments are presented as DHFEs (DH decision matrix (DHDM) $[DHFE]_{4\times4} = [d_{ii'}]_{4\times4}$) given in Table 2). The assessment about the real estate agent $\mathscr{A}_1$ for criteria $\mathscr{C}_1$ is of the form of DHFE $\langle(0.2, 0.3), (0.3, 0.5, 0.6)\rangle$, indicating that two of the three analysts gives the same membership value to $\mathscr{A}_1$ under the criteria $\mathscr{C}_1$ to be 0.2, and the remaining one gives the value 0.3. Whereas the three experts give the non-membership value to $\mathscr{A}_1$ under the criteria $\mathscr{C}_1$ as $0.3, 0.5$ and $0.6$ respectively.

**Step-2.** In this illustration all the criteria $\mathscr{C}_i(i = 1, 2, \cdots, 4)$ belong to beneficial type index of criteria. Thus there will be no changing in DHDM $[DHFE]_{4\times4} = [d_{ii'}]_{4\times4}$.

**Step-3.** Experts refer to the realistic assessment data, widely recognized among real estate businesses, as the reference set, as presented in Table 3.

**Step-4.** Now we converted DHDM and reference set to NWDHFSs using Eqs. (2)–(6). We obtained NWDHF decision matrix (NWDHFDMs) in Tables 4 and 5.

**Step-5.** Utilizing Eqs. (8)–(15), we calculated the mean values of NWDHFEs provided in Table 4 and encompassed them into Table 6. Similarly using Eqs. (8)–(15), we calculated the mean values of NWDHFEs provided in Table 5 and encompassed them into Table 7.

**Step-5.** We proceeded by computing the CCs $\Delta_{NWDH_1}$ by applying Definition 7 and given as follows;

$$\Delta_{NWDH1}(\mathscr{A}_1, \mathscr{Y}^*) = -0.3546, \quad \Delta_{NWDH1}(\mathscr{A}_2, \mathscr{Y}^*) = 0.1657,$$

**Table 2  DHFSs based decision matrix.**

| C/A | $\mathscr{A}_1$ | $\mathscr{A}_2$ | $\mathscr{A}_3$ | $\mathscr{A}_4$ |
|---|---|---|---|---|
| $\mathscr{C}_1$ | $\langle(0.2,0.3),(0.3,0.5,0.6)\rangle$ | $\langle(0.1,0.2,0.3),(0.4,0.6)\rangle$ | $\langle(0.3,0.4),(0.3,0.4,0.5)\rangle$ | $\langle(0.1,0.2,0.3),(0.4,0.5,0.6)\rangle$ |
| $\mathscr{C}_2$ | $\langle(0.3,0.5),(0.4,0.5,0.6)\rangle$ | $\langle(0.3,0.4,0.5),(0.2,0.4)\rangle$ | $\langle(0.2,0.5),(0.3,0.6)\rangle$ | $\langle(0.1,0.4),(0.3,0.4,0.5)\rangle$ |
| $\mathscr{C}_3$ | $\langle(0.2,0.3,0.4),(0.1,0.4)\rangle$ | $\langle(0.2,0.4,0.6),(0.3,0.5)\rangle$ | $\langle(0.1,0.2,0.4),(0.5,0.7)\rangle$ | $\langle(0.1,0.3),(0.4,0.6)\rangle$ |
| $\mathscr{C}_4$ | $\langle(0.4,0.6,0.8),(0.2,0.5)\rangle$ | $\langle(0.1,0.4,0.5),(0.2,0.5)\rangle$ | $\langle(0.3,0.4,0.5),(0.2,0.4,0.5)\rangle$ | $\langle(0.2,0.4,0.5),(0.4,0.5)\rangle$ |

**Table 3  Reference set $\mathscr{B}^*$.**

| $\mathscr{C}/\mathscr{B}^*$ | $\mathscr{B}^*$ |
|---|---|
| $\mathscr{C}_1$ | $\langle(0.3,0.5,0.7),(0.4,0.5)\rangle$ |
| $\mathscr{C}_2$ | $\langle(0.2,0.6),(0.3,0.4,0.5)\rangle$ |
| $\mathscr{C}_3$ | $\langle(0.1,0.5),(0.4,0.6)\rangle$ |
| $\mathscr{C}_4$ | $\langle(0.1,0.2,0.4),(0.3,0.4,0.5)\rangle$ |

$\Delta_{NWDH1}(\mathscr{A}_3,\mathscr{Y}^*)=0.8032$, $\Delta_{NWDH1}(\mathscr{A}_4,\mathscr{Y}^*)=0.7476$.
Further we calculated the WCCs $\Delta_{NWDH_{\varsigma}2}$ using Eq. (9) and presented the results as follows.

$\Delta_{NWDH_{\varsigma1}}(\mathscr{A}_1,\mathscr{Y}^*)=-0.3664$, $\Delta_{NWDH_{\varsigma1}}(\mathscr{A}_2,\mathscr{Y}^*)=0.1752$,
$\Delta_{NWDH_{\varsigma1}}(\mathscr{A}_3,\mathscr{Y}^*)=0.8009$, $\Delta_{NWDH_{\varsigma1}}(\mathscr{A}_4,\mathscr{Y}^*)=0.7590$

**Step-6.** Table 8 illustrates the final ranking based on the computed values of CCs on NWDHFSs in Step-5.

**Step-7.** The most optimal real estates agent in our case study using $\Delta_{NWDH_1}$ is $\mathscr{A}_3$. The most optimal real estates agent in our case study using $\Delta_{NWDH_{\varphi1}}$ is $\mathscr{A}_3$.

In addition, Table 8 displays the weighted CCs on NWDHFSs $\Delta_{NWDH_{\varsigma1}}$ or $\Delta_{NWDH_{\varsigma2}}$ computed using the weighted vector $\varsigma=\left\{\frac{0.4}{\varsigma_1},\frac{0.3}{\varsigma_2},\frac{0.2}{\varsigma_3},\frac{0.1}{\varsigma_4}\right\}$.

## SENSITIVITY AND COMPARATIVE ANALYSIS

In the illustration presented in 'Application of the Method for the Selection of Real Estate Agent', we calculated four categories of correlation coefficients: $\Delta_{NWDH_1}$, $\Delta_{NWDH_2}$, $\Delta_{NWDH_{\varsigma1}}$ and $\Delta_{NWDH_{\varsigma2}}$. The ordering illustrated in Table 8 differs for CCs $\Delta_{NWDH_1}$ and $\Delta_{NWDH_2}$; we utilize the maximum of correlations in the denominators of $\Delta_{NWDH_2}$, as this method disregards one value in the denominator. The methodology of CCs $\Delta_{NWDH_1}$ is more appropriate, as utilizing the product of two correlations in the denominator is justifiable due to the absence of neglect towards the values in the denominator.

Furthermore, $\Delta_{NWDH_{\varsigma1}}$ and $\Delta_{NWDH_{\varsigma2}}$ are the weighted CCs on NWDHFSs in which we used a weighted vector over criteria. Although the ranking on $\Delta_{NWDH_1}$ and $\Delta_{NWDH_{\varsigma1}}$ as depicted in Table 8 is similar, the CC $\Delta_{NWDH_{\varsigma1}}$ is important when preference related to criteria is necessary. Therefore their ranking may or may not be similar to each other.

**Table 4  NWDHFSs based decision matrix.**

| NWDHFS | $\mathcal{A}_1$ |
|---|---|
| $\mathscr{C}_1$ | $\langle(0.2, 0.3), (0.3, 0.5, 0.6)\{(0.16987, 0.1939, 0.2303), (0.2697, 0.2939, 0.3303)\}\{(0.249, 0.2891, 0.351), (0.3797, 0.4742, 0.6203), (0.5296, 0.5849, 0.6704)\}\rangle$ |
| $\mathscr{C}_2$ | $\langle(0.3, 0.5), (0.4, 0.5, 0.6)\{(0.2393, 0.2848, 0.3607), (0.4393, 0.4848, 0.5607)\}\{(0.3615, 0.4, 0.4385), (0.4184, 0.5, 0.5816), (0.5615, 0.6, 0.6385)\}\rangle$ |
| $\mathscr{C}_3$ | $\langle(0.2, 0.3, 0.4), (0.1, 0.4)\{(0.1615, 0.1914, 0.2385), (0.2184, 0.2819, 0.3816)(0.3615, 0.3914, 0.4385)\}\{(0.009, 0.0454, 0.191), (0.309, 0.3454, 0.491)\}\rangle$ |
| $\mathscr{C}_4$ | $\langle(0.4, 0.6, 0.8), (0.2, 0.5)\{(0.3229, 0.4171, 0.4771), (0.4367, 0.6363, 0.7633), (0.7229, 0.8171, 0.8771)\}\{(0.109, 0.161, 0.291), (0.409, 0.461, 0.591)\}\rangle$ |

| NWDHFS | $\mathcal{A}_2$ |
|---|---|
| $\mathscr{C}_1$ | $\langle(0.1, 0.2, 0.3), (0.4, 0.6)\{(0.0615, 0.0872, 0.1385), (0.1184, 0.1728, 0.2816), (0.2615, 0.2872, 0.3385)\}, \{(0.3393, 0.4, 0.4607), (0.5393, 0.6, 0.6607)\}\rangle$ |
| $\mathscr{C}_2$ | $\langle(0.3, 0.4, 0.5), (0.2, 0.4)\{(0.2615, 0.2936, 0.3385), (0.3184, 0.3864, 0.4816), (0.4615, 0.4936, 0.5385)\}, \{(0.1393, 0.1798, 0.2607), (0.3393, 0.3798, 0.4607)\}\rangle$ |
| $\mathscr{C}_3$ | $\langle(0.2, 0.4, 0.6), (0.3, 0.5)\{(0.1229, 0.1743, 0.2771), (0.2367, 0.3456, 0.5633), (0.5229, 0.5743, 0.6771)\}, \{(0.2393, 0.2848, 0.3607), (0.4393, 0.4848, 0.5607)\}\rangle$ |
| $\mathscr{C}_4$ | $\langle(0.1, 0.4, 0.5), (0.2, 0.5)\{(0.0337, 0.0735, 0.1663), (0.2426, 0.3370, 0.5574), (0.3949, 0.4580, 0.6051)\}, \{(0.109, 0.161, 0.291), (0.409, 0.461, 0.591)\}\rangle$ |

| NWDHFS | $\mathcal{A}_3$ |
|---|---|
| $\mathscr{C}_1$ | $\langle(0.3, 0.4), (0.3, 0.4, 0.5)\{(0.2697, 0.2957, 0.3303), (0.3697, 0.3957, 0.4303)\}, \{(0.2615, 0.2936, 0.3385), (0.3184, 0.3864, 0.4816), (0.4615, 0.4936, 0.5385)\}\rangle$ |
| $\mathscr{C}_2$ | $\langle(0.2, 0.5), (0.3, 0.6)\{(0.109, 0.161, 0.291), (0.409, 0.461, 0.591)\}, \{(0.209, 0.2697, 0.391), (0.509, 0.5697, 0.691)\}\rangle$ |
| $\mathscr{C}_3$ | $\langle(0.1, 0.2, 0.4), (0.5, 0.7)\{(0.0296, 0.0698, 0.1704), (0.0797, 0.1484, 0.3203), (0.349, 0.3781, 0.451)\}, \{(0.4393, 0.5101, 0.5607), (0.6393, 0.7101, 0.7607)\}\rangle$ |
| $\mathscr{C}_4$ | $\langle(0.3, 0.4, 0.5), (0.2, 0.4, 0.5)\{(0.2615, 0.2936, 0.3385), (0.3184, 0.3864, 0.4816), (0.4615, 0.4936, 0.5385)\}; \{(0.149, 0.1861, 0.251), (0.2797, 0.3682, 0.5203), (0.4296, 0.4808, 0.5704)\}\rangle$ |

| NWDHFS | $\mathcal{A}_4$ |
|---|---|
| $\mathscr{C}_1$ | $\langle(0.1, 0.2, 0.3), (0.4, 0.5, 0.6)\{(0.0615, 0.0872, 0.1385), (0.1184, 0.1728, 0.2816), (0.2615, 0.2872, 0.3385)\}, \{(0.3615, 0.4, 0.4385), (0.4184, 0.5, 0.5816), (0.5615, 0.6, 0.6385)\}\rangle$ |
| $\mathscr{C}_2$ | $\langle(0.1, 0.4), (0.3, 0.4, 0.5)\{(0.009, 0.1, 0.191), (0.309, 0.4, 0.491)\}, \{(0.2615, 0.2936, 0.3385), (0.3184, 0.3864, 0.4816), (0.4615, 0.4936, 0.5385)\}\rangle$ |
| $\mathscr{C}_3$ | $\langle(0.1, 0.3), (0.4, 0.6)\{(0.0393, 0.0697, 0.1607), (0.2393, 0.2697, 0.3607)\}; \{(0.3393, 0.4, 0.4607), (0.5393, 0.6, 0.6607)\}\rangle$ |
| $\mathscr{C}_4$ | $\langle(0.2, 0.4, 0.5), (0.4, 0.5)\{(0.149, 0.1861, 0.251), (0.2797, 0.3682, 0.5203), (0.4296, 0.4808, 0.5704)\}, \{(0.3697, 0.3966, 0.4303), (0.4697, 0.4966, 0.5303)\}\rangle$ |

**Table 5  NWDHFS for reference set $\mathcal{Y}^*$.**

| NWDHFS | $\mathcal{Y}^*$ |
|---|---|
| $\mathscr{C}_1$ | $\langle(0.3, 0.5, 0.7), (0.4, 0.5)\{(0.2229, 0.3, 0.3771), (0.3367, 0.5, 0.6633), (0.6229, 0.7, 0.7771)\}\{(0.3697, 0.3966, 0.4303), (0.4697, 0.4966, 0.5303)\}\rangle$ |
| $\mathscr{C}_2$ | $\langle(0.2, 0.6), (0.3, 0.4, 0.5)\{(0.0787, 0.1394, 0.3213), (0.4787, 0.5394, 0.7213)\}\{(0.2615, 0.2936, 0.3385), (0.3184, 0.3864, 0.4816), (0.4615, 0.4936, 0.5385)\}\rangle$ |
| $\mathscr{C}_3$ | $\langle(0.1, 0.5), (0.4, 0.6)\{(0, 0.0191, 0.2213), (0.3787, 0.4191, 0.6213)\}\{(0.3393, 0.4, 0.4607), (0.5393, 0.6, 0.6607)\}\rangle$ |
| $\mathscr{C}_4$ | $\langle(0.1, 0.2, 0.4), (0.3, 0.4, 0.5)\{(0.0296, 0.0698, 0.1704), (0.0797, 0.1484, 0.3203), (0.349, 0.3781, 0.451)\}\{(0.2615, 0.2936, 0.3385), (0.3184, 0.3864, 0.4816), (0.4616, 0.4936, 0.5385)\}\rangle$ |

**Table 6 Mean values of NWDHFEs from Table 4.**

| $\mathscr{C}$ | $\langle \bar{\aleph}_{\mathscr{A}_1}, \bar{\Upsilon}_{\mathscr{A}_1}, \bar{\Gamma}_{\mathscr{A}_1}(\aleph), \bar{\Psi}_{\mathscr{A}_1}(\Upsilon) \rangle$ | $\langle \bar{\aleph}_{\mathscr{A}_2}, \bar{\Upsilon}_{\mathscr{A}_2}, \bar{\Gamma}_{\mathscr{A}_2}(\aleph), \bar{\Psi}_{\mathscr{A}_2}(\Upsilon) \rangle$ | $\langle \bar{\aleph}_{\mathscr{A}_3}, \bar{\Upsilon}_{\mathscr{A}_3}, \bar{\Gamma}_{\mathscr{A}_3}(\aleph), \bar{\Psi}_{\mathscr{A}_3}(\Upsilon) \rangle$ | $\langle \bar{\aleph}_{\mathscr{A}_4}, \bar{\Upsilon}_{\mathscr{A}_4}, \bar{\Gamma}_{\mathscr{A}_4}(\aleph), \bar{\Psi}_{\mathscr{A}_4}(\Upsilon) \rangle$ |
|---|---|---|---|---|
| $\mathscr{C}_1$ | $\langle 0.2500, 0.4667, 0.2480, 0.4609 \rangle$ | $\langle 0.2000, 0.5000, 0.1941, 0.5 \rangle$ | $\langle 0.3500, 0.4000, 0.3486, 0.3971 \rangle$ | $\langle 0.2000, 0.5000, 0.1941, 0.5000 \rangle$ |
| $\mathscr{C}_2$ | $\langle 0.4000, 0.5000, 0.3949, 0.5000 \rangle$ | $\langle 0.4000, 0.3000, 0.3971, 0.2933 \rangle$ | $\langle 0.3500, 0.4500, 0.3370, 0.4399 \rangle$ | $\langle 0.2500, 0.4000, 0.2318, 0.3971 \rangle$ |
| $\mathscr{C}_3$ | $\langle 0.3000, 0.2500, 0.2961, 0.2318 \rangle$ | $\langle 0.3667, 0.4000, 0.3882, 0.3949 \rangle$ | $\langle 0.2333, 0.6000, 0.2218, 0.6043 \rangle$ | $\langle 0.2000, 0.5000, 0.1899, 0.5000 \rangle$ |
| $\mathscr{C}_4$ | $\langle 0.6000, 0.3500, 0.6078, 0.3370 \rangle$ | $\langle 0.3333, 0.3500, 0.3187, 0.337 \rangle$ | $\langle 0.4000, 0.3667, 0.3971, 0.3593 \rangle$ | $\langle 0.3667, 0.4500, 0.3595, 0.4489 \rangle$ |

**Table 7 Mean values of NWDHFEs from Table 5.**

| $\mathscr{Y}^*$ | $\langle \bar{\aleph}_{\mathscr{Y}^*}, \bar{\Upsilon}_{\mathscr{Y}^*}, \bar{\Gamma}_{\mathscr{Y}^*}(\aleph), \bar{\Psi}_{\mathscr{Y}^*}(\Upsilon) \rangle$ |
|---|---|
| $\mathscr{C}_1$ | $\langle 0.5000, 0.4500, 0.5000, 0.4489 \rangle$ |
| $\mathscr{C}_2$ | $\langle 0.4000, 0.4000, 0.3798, 0.3971 \rangle$ |
| $\mathscr{C}_3$ | $\langle 0.3000, 0.5000, 0.2766, 0.5000 \rangle$ |
| $\mathscr{C}_4$ | $\langle 0.2333, 0.4000, 0.2218, 0.3971 \rangle$ |

**Table 8 The ranking of real estates agents using CCs NWDHFSs.**

| Correlation coefficients/Distance measure | Obtained values | Ranking |
|---|---|---|
| $\Delta_{NWDH_1}$ | $\Delta_{NWDH_1}(\mathscr{A}_1, \mathscr{Y}^*) = 0.3899$, $\Delta_{NWDH_1}(\mathscr{A}_2, \mathscr{Y}^*) = 0.7650$, $\Delta_{NWDH_1}(\mathscr{A}_3, \mathscr{Y}^*) = 0.8363$, $\Delta_{NWDH1}(\mathscr{A}_4, \mathscr{Y}^*) = 0.9126$ | $\mathscr{A}_1 \leq \mathscr{A}_2 \leq \mathscr{A}_3 \leq \mathscr{A}_4$ |
| $\Delta_{NWDH_2}$ | $\Delta_{NWDH_2}(\mathscr{A}_1, \mathscr{Y}^*) = 0.2966$, $\Delta_{NWDH_2}(\mathscr{A}_2, \mathscr{Y}^*) = 0.7122$, $\Delta_{NWDH_2}(\mathscr{A}_3, \mathscr{Y}^*) = 0.5597$, $\Delta_{NWDH_2}(\mathscr{A}_4, \mathscr{Y}^*) = 0.6315.$ | $\mathscr{A}_1 \leq \mathscr{A}_3 \leq \mathscr{A}_4 \leq \mathscr{A}_2$ |
| $\Delta_{NWDH_{\varsigma 1}}$ | $\Delta_{NWDH_{\varsigma 1}}(\mathscr{A}_1, \mathscr{Y}^*) = 0.3656$, $\Delta_{NWDH_{\varsigma 1}}(\mathscr{A}_2, \mathscr{Y}^*) = 0.7140$, $\Delta_{NWDH_{\varsigma 1}}(\mathscr{A}_3, \mathscr{Y}^*) = 0.8503$, $\Delta_{NWDH_{\varsigma 1}}(\mathscr{A}_4, \mathscr{Y}^*) = 0.8985$ | $\mathscr{A}_1 \leq \mathscr{A}_2 \leq \mathscr{A}_3 \leq \mathscr{A}_4$ |
| $\Delta_{NWDH_{\varsigma 2}}$ | $\Delta_{NWDH_{\varsigma 2}}(\mathscr{A}_1, \mathscr{Y}^*) = 0.2486$, $\Delta_{NWDH_{\varsigma 2}}(\mathscr{A}_2, \mathscr{Y}^*) = 0.5817$, $\Delta_{NWDH_{\varsigma 2}}(\mathscr{A}_3, \mathscr{Y}^*) = 0.5950$, $\Delta_{NWDH_{\varsigma 2}}(\mathscr{A}_4, \mathscr{Y}^*) = 0.6774$ | $\mathscr{A}_1 \leq \mathscr{A}_2 \leq \mathscr{A}_3 \leq \mathscr{A}_4$ |
| $\rho_{NW_1}$ (*Wang et al., 2024*) | $\rho_{NW_1}(\mathscr{A}_1, \mathscr{Y}^*) = 0.09824$, $\rho_{NW_1}(\mathscr{A}_2, \mathscr{Y}^*) = -0.3207$, $\rho_{NW_1}(\mathscr{A}_3, \mathscr{Y}^*) = 0.9735$, $\rho_{NW_1}(\mathscr{A}_4, \mathscr{Y}^*) = 0.9454$ | $\mathscr{A}_2 < \mathscr{A}_1 < \mathscr{A}_4 < \mathscr{A}_3$ |
| $\rho_{NW_{\omega 1}}$ (*Wang et al., 2024*) | $\rho_{NW_{\omega 1}}(\mathscr{A}_1, \mathscr{Y}^*) = 0.1793$, $\rho_{NW_{\omega 1}}(\mathscr{A}_2, \mathscr{Y}^*) = -0.3924$, $\rho_{NW_{\omega 1}}(\mathscr{A}_3, \mathscr{Y}^*) = 0.9830$ $\rho_{NW_{\omega 1}}(\mathscr{A}_4, \mathscr{Y}^*) = 0.9475$ | $\mathscr{A}_2 < \mathscr{A}_1 < \mathscr{A}_4 < \mathscr{A}_3$ |
| $\rho_{NW_2}$ (*Wang et al., 2024*) | $\rho_{NW_2}(\mathscr{A}_1, \mathscr{Y}^*) = 0.09613$, $\rho_{NW_2}(\mathscr{A}_2, \mathscr{Y}^*) = -0.2542$, $\rho_{NW_2}(\mathscr{A}_3, \mathscr{Y}^*) = 0.5194$, $\rho_{NW_2}(\mathscr{A}_4, \mathscr{Y}^*) = 0.6313$ | $\mathscr{A}_2 < \mathscr{A}_1 < \mathscr{A}_3 < \mathscr{A}_4$ |
| $\rho_{NW_{\omega 2}}$ (*Wang et al., 2024*) | $\rho_{NW_{\omega 2}}(\mathscr{A}_1, \mathscr{Y}^*) = 0.1639$, $\rho_{NW_{\omega 2}}(\mathscr{A}_2, \mathscr{Y}^*) = -0.27026$, $\rho_{NW_{\omega 2}}(\mathscr{A}_3, \mathscr{Y}^*) = 0.5544$, $\rho_{NW_{\omega 2}}(\mathscr{A}_4, \mathscr{Y}^*) = 0.6764$ | $\mathscr{A}_2 < \mathscr{A}_1 < \mathscr{A}_3 < \mathscr{A}_4$ |
| $CC_1$ (*Meng, Xu & Wang, 2019*) | $CC_1(\mathscr{A}_1, \mathscr{Y}^*) = 0.9461$, $CC_1(\mathscr{A}_2, \mathscr{Y}^*) = 0.9672$, $CC_1(\mathscr{A}_3, \mathscr{Y}^*) = 1.0000$, $CC_1(\mathscr{A}_4, \mathscr{Y}^*) = 0.9671$ | $\mathscr{A}_1 < \mathscr{A}_4 < \mathscr{A}_2 < \mathscr{A}_3$ |
| $CC_2$ (*Meng, Xu & Wang, 2019*) | $CC_2(\mathscr{A}_1, \mathscr{Y}^*) = 0.8776$, $CC_2(\mathscr{A}_2, \mathscr{Y}^*) = 0.8779$, $CC_1(\mathscr{A}_3, \mathscr{Y}^*) = 1.0000$, $CC_2(\mathscr{A}_4, \mathscr{Y}^*) = 0.8845$ | $\mathscr{A}_1 < \mathscr{A}_2 < \mathscr{A}_4 < \mathscr{A}_3$ |
| Distance measure $\chi_1$ (*Boulaaras et al., 2024*) | $\chi_1(\mathscr{A}_1, \mathscr{Y}^*) = 0.1110$, $\chi_1(\mathscr{A}_2, \mathscr{Y}^*) = 0.2042$, $\chi_1(\mathscr{A}_3, \mathscr{Y}^*) = 0.0685$, $\chi_1(\mathscr{A}_4, \mathscr{Y}^*) = 0.0673$ | $\mathscr{A}_2 < \mathscr{A}_1 < \mathscr{A}_3 < \mathscr{A}_4$ |

**Table 9 HFS based decision matrix.**

| $\mathscr{C}/\mathscr{A}$ | $\mathscr{A}_1$ | $\mathscr{A}_2$ | $\mathscr{A}_3$ | $\mathscr{A}_4$ |
|---|---|---|---|---|
| $\mathscr{C}_1$ | $(0.2, 0.3)$ | $(0.1, 0.2, 0.3)$ | $(0.3, 0.4)$ | $(0.1, 0.2, 0.3)$ |
| $\mathscr{C}_2$ | $(0.3, 0.5)$ | $(0.3, 0.4, 0.5)$ | $(0.2, 0.5)$ | $(0.1, 0.4)$ |
| $\mathscr{C}_3$ | $(0.2, 0.3, 0.4)$ | $(0.2, 0.4, 0.5)$ | $(0.1, 0.2, 0.4)$ | $(0.1, 0.3)$ |
| $\mathscr{C}_4$ | $(0.4, 0.6, 0.8)$ | $(0.1, 0.4, 0.5)$ | $(0.3, 0.4, 0.5)$ | $(0.2, 0.4, 0.5)$ |

**Table 10 Reference set $\mathscr{Y}^*$.**

| $\mathscr{C}/\mathscr{Y}^*$ | $\mathscr{Y}^*$ |
|---|---|
| $\mathscr{C}_1$ | $(0.3, 0.5, 0.7)$ |
| $\mathscr{C}_2$ | $(0.2, 0.6)$ |
| $\mathscr{C}_3$ | $(0.1, 0.5)$ |
| $\mathscr{C}_4$ | $(0.1, 0.2, 0.4)$ |

Next we take into count method on NWHFSs presented by *Wang et al. (2024)*. In order to compare this method with method of *Wang et al. (2024)*, we consider only membership values from Tables 2 and 3. Thus the resultant tables are in the form of HFEs and expressed in Tables 9 and 10. Utilizing the method of *Wang et al. (2024)* we obtained the following values of CCs $\rho_{NW_1}$ from Tables 9, 10; $\rho_{NW_1}(\mathscr{A}_1, \mathscr{Y}^*) = 0.09824$, $\rho_{NW_1}(\mathscr{A}_2, \mathscr{Y}^*) = -0.3207$, $\rho_{NW_1}(\mathscr{A}_3, \mathscr{Y}^*) = 0.9735$ and $\rho_{NW_1}(\mathscr{A}_4, \mathscr{Y}^*) = 0.9454$. The related rating as $\mathscr{A}_2 < \mathscr{A}_1 < \mathscr{A}_4 < \mathscr{A}_3$. Similarly, utilizing weighted CCs $\rho_{NW_{\omega1}}$ by *Wang et al. (2024)* we obtained the following values from Tables 9, 10; $\rho_{NW_{\omega1}}(\mathscr{A}_1, \mathscr{Y}^*) = 0.1793$, $\rho_{NW_{\omega1}}(\mathscr{A}_2, \mathscr{Y}^*) = -0.3924$, $\rho_{NW_{\omega1}}(\mathscr{A}_3, \mathscr{Y}^*) = 0.9830$ and $\rho_{NW_{\omega1}}(\mathscr{A}_4, \mathscr{Y}^*) = 0.9475$. The related rating as $\mathscr{A}_2 < \mathscr{A}_1 < \mathscr{A}_4 < \mathscr{A}_3$. Next $\rho_{NW_2}$ and $\rho_{NW_{\omega2}}$ from work of *Wang et al. (2024)* are calculated as $\rho_{NW_2}(\mathscr{A}_1, \mathscr{Y}^*) = 0.09613$, $\rho_{NW_2}(\mathscr{A}_2, \mathscr{Y}^*) = -0.2542$, $\rho_{NW_2}(\mathscr{A}_3, \mathscr{Y}^*) = 0.5194$, $\rho_{NW_2}(\mathscr{A}_4, \mathscr{Y}^*) = 0.6313$ and $\rho_{NW_{\omega2}}(\mathscr{A}_1, \mathscr{Y}^*) = 0.1639$, $\rho_{NW_{\omega2}}(\mathscr{A}_2, \mathscr{Y}^*) = -0.27026$, $\rho_{NW_{\omega2}}(\mathscr{A}_3, \mathscr{Y}^*) = 0.5544$, $\rho_{NW_{\omega2}}(\mathscr{A}_4, \mathscr{Y}^*) = 0.6764$. The ranking of both the CCs $\rho_{NW_2}$ and $\rho_{NW_{\omega2}}$ is as follows; $\mathscr{A}_2 < \mathscr{A}_1 < \mathscr{A}_4 < \mathscr{A}_3$. The Fig. 2 presents a visual representation of a study of the orders of alternative values that were produced from the suggested method and *Wang et al. (2024)*.

The results acquired from the study of *Wang et al. (2024)* deviate from the recommended method. The technical rationale is that the method proposed by *Wang et al. (2024)* lacks dual values for the evaluations in HFSs. The duality is a critical aspect in complex real-world problems. The proposed method processes information in the form of NWDHFSs, hence demonstrating superior performance in duality cases compared to the approach of *Wang et al. (2024)*.

Subsequently, we evaluated our approach in comparison to the method proposed by *Meng, Xu & Wang (2019)* utilizing Tables 2, 3 of DHFSs. By calculating CCs $CC_1$ and $CC_2$, we derived the following values, respectively: $CC_1(\mathscr{A}_1, \mathscr{Y}^*) = 0.9461$, $CC_1(\mathscr{A}_2, \mathscr{Y}^*) = 0.9672$, $CC_1(\mathscr{A}_3, \mathscr{Y}^*) = 1.0000$, $CC_1(\mathscr{A}_4, \mathscr{Y}^*) = 0.9671$ and $CC_2(\mathscr{A}_1, \mathscr{Y}^*) = 0.0.8776$, $CC_2(\mathscr{A}_2, \mathscr{Y}^*) = 0.8779$, $CC_1(\mathscr{A}_3, \mathscr{Y}^*) = 1.0000$, $CC_2(\mathscr{A}_4, \mathscr{Y}^*) = 0.8845$. The corresponding rankings for $CC_1$ and $CC_2$ are $CC_1$ and $CC_2$ is

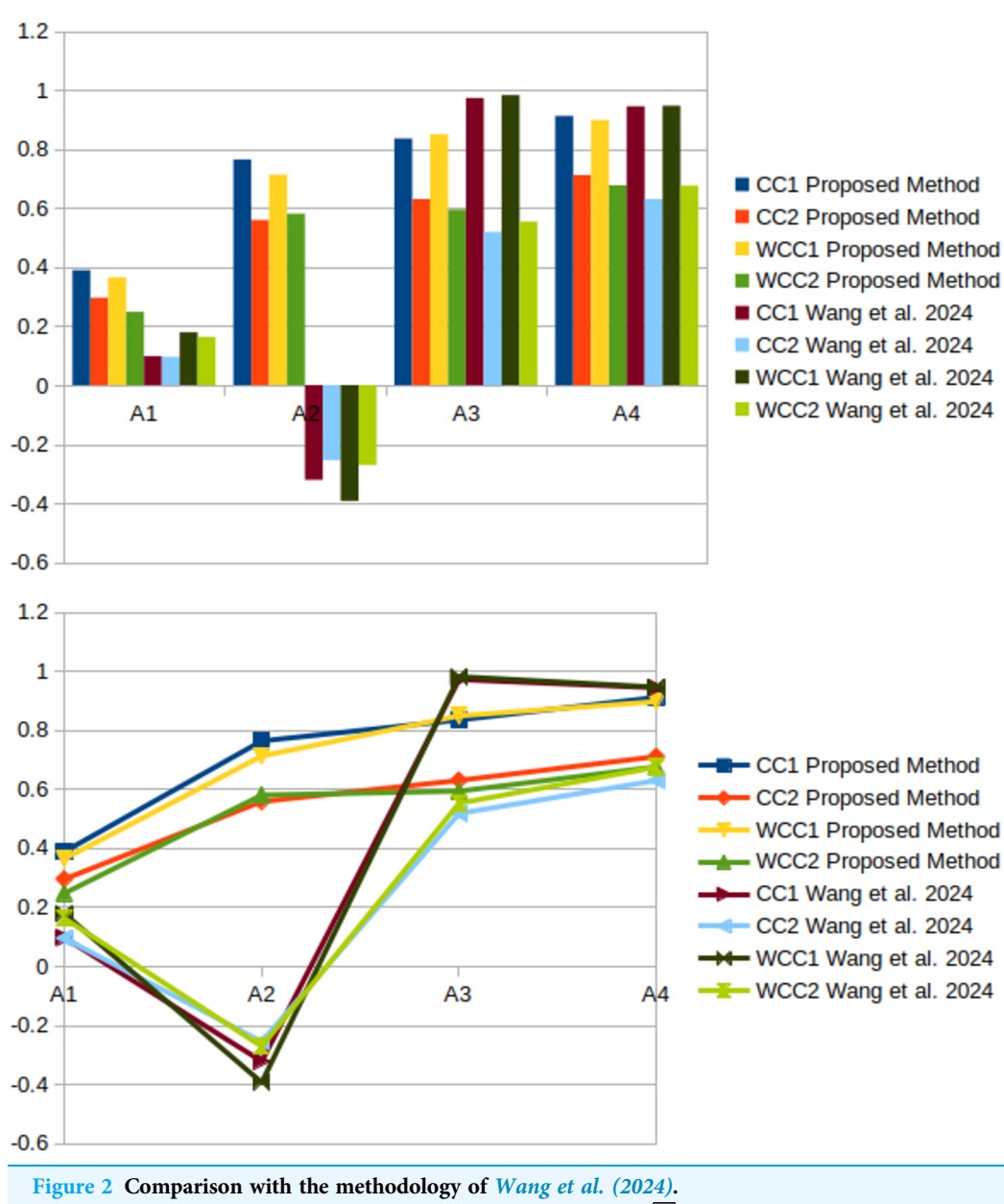

**Figure 2 Comparison with the methodology of *Wang et al. (2024)*.**

$\mathscr{A}_1 < \mathscr{A}_4 < \mathscr{A}_2 < \mathscr{A}_3$ and $\mathscr{A}_1 < \mathscr{A}_2 < \mathscr{A}_4 < \mathscr{A}_3$, respectively. The Fig. 3 presents a visual representation of a study of the orders of alternative values that were produced from the suggested method and *Meng, Xu & Wang (2019)*.

It is possible to check the order of alternatives by utilizing the approach described by *Meng, Xu & Wang (2019)*, which is distinct from the way that is being suggested in this study. This is due to the fact that *Meng, Xu & Wang (2019)* solely relied on DHFEs in their framework, despite the fact that our method changed DHFSs to NWDHFSs. This is due to

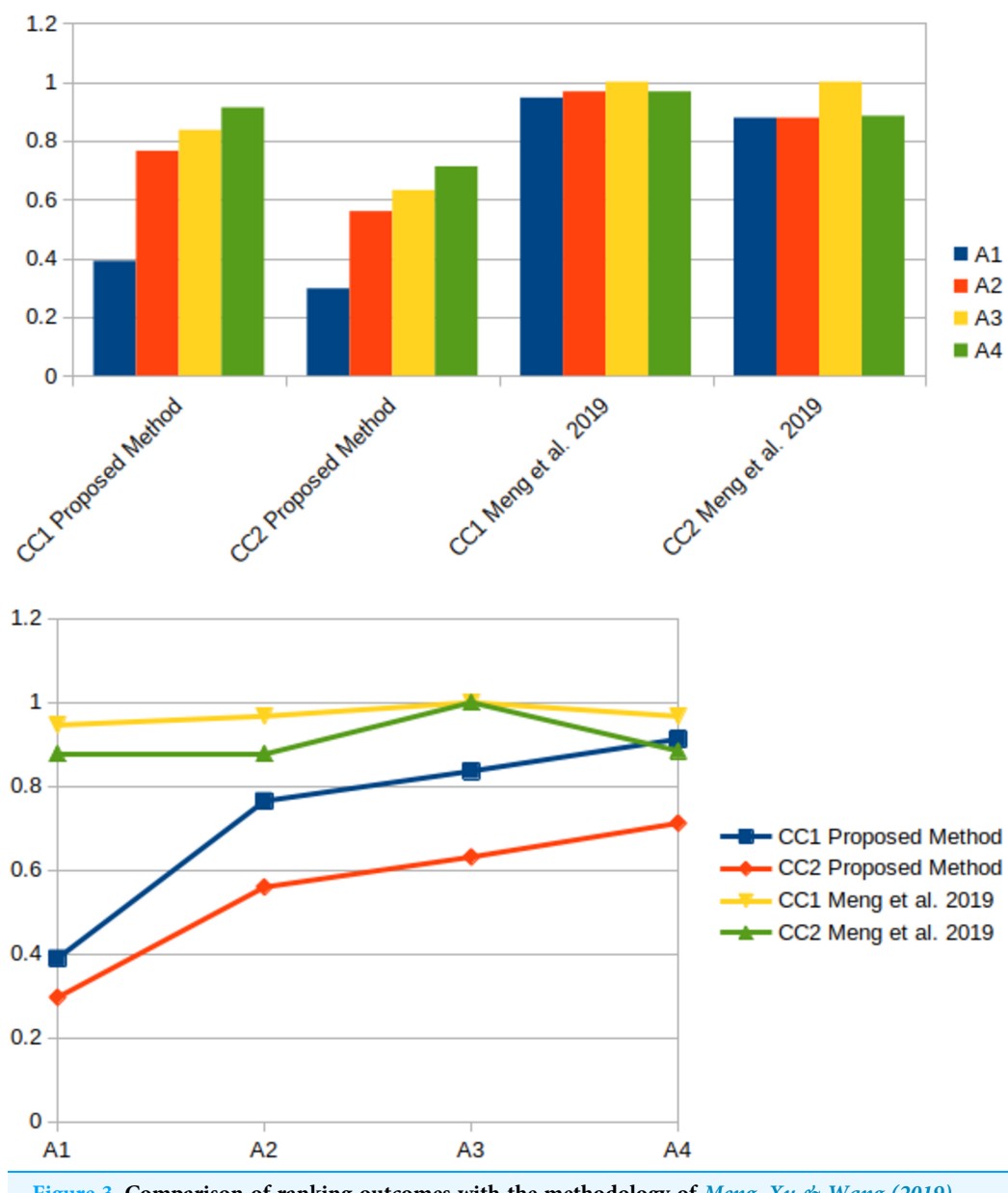

**Figure 3 Comparison of ranking outcomes with the methodology of *Meng, Xu & Wang (2019)*.**

the fact that the wiggly values are composed of triple means type values of HFEs, which allow us to overcome impressions more realistically.

The proposed work is contrasted with the work of *Boulaaras et al. (2024)* in order to elaborate the comparative and sensitivity analysis. The following outcomes were obtained by using the distance measure defined by *Boulaaras et al. (2024)* on Tables 2 and 3; $\chi_1(\mathscr{A}_1, \mathscr{Y}^*) = 0.1110$, $\chi_1(\mathscr{A}_2, \mathscr{Y}^*) = 0.2042$, $\chi_1(\mathscr{A}_3, \mathscr{Y}^*) = 0.0685$, $\chi_1(\mathscr{A}_4, \mathscr{Y}^*) = 0.0673$. The ranking can be checked as given by $\mathscr{A}_4 < \mathscr{A}_3 < \mathscr{A}_1 < \mathscr{A}_2$.

Clearly, the optimal result is $\mathscr{A}_4$, it has the minimum distance with $\mathscr{Y}^*$. This ranking varies from the recommended methodology as distance measures neglect internal means within memberships in HFEs.

Table 8 illustrates that various approaches may yield divergent ranking patterns and distinct ideal selections; therefore, decision-makers must select an appropriate decision-making method based on actual requirements prior to making decisions. We generally advise the DMs to implement the new method. The primary advantages of the new procedures, in comparison to prior techniques, are:

1. NWDHFCCs can address the interactive properties among elements within a set;
2. NWDHFCCs enable the resolution of duality and hesitation factors inherent in NWDHFSs.
3. Novel methodologies can address scenarios where fuzzy measures are partially understood and options are varied;
4. Innovative approaches broaden the range of choice values and the confidence levels of DMs.

## CONCLUSIONS

In the MCDM process, the degree of hesitation, MGs and NMGs are influenced not just by the format or magnitude of the DM's weighted input but also by the uncertain and subjective feelings of the DMs. According to constrained logic, an inherent characteristic of humanity, individuals' perceptions of any certain numerical value are likely to exist within a spectrum. The actual configuration of the spectrum may be interval, triangle, trapezoidal, or other forms, which corresponds to the initial assessment data provided by the DMs.

In NWDHFSs the NWDHFE is a visualization that identifies the standard oscillatory spectrum for each value in a HFE. To explore the enormous uncertainty of hesitant fuzzy information, we consider the problem of CCs on NWDHFSs. We have presented a multi-criteria decision-making technique and associated algorithms based on these CCs. Through the examination of a real estate case study, we have derived appropriate assessments of real estate agents for real estate firms utilizing NWDHFSs. We have analyzed the techniques and results of our approach in comparison to several existing techniques.

In complicated systems, numerous issues arise when hesitant fuzzy information or DHFS based information prove to be inadequate due to the disconnection among grades. Consequently, under these conditions, NWDHFSs and CCs on NWDHFSs can serve as an essential instrument for application. In future work, we will address proposed CCs on NWDHFSs in clustering analysis, medical diagnoses, image segmentation and recommendation systems. Moreover, we will define distance and similarity measure on NWDHFSs, and also we will define CCs on probabilistic NWDHFSs.

### Funding

This article was supported by the NRPU-HEC Pakistan Project Number 14662 and the joint project PSF(PSF-NSFC/JSEP/ENG/AJKUKAJK/01)-NSFC (12211540710). The funders had no role in study design, data collection and analysis, decision to publish, or preparation of the manuscript.

### Grant Disclosures

The following grant information was disclosed by the authors:
NRPU-HEC Pakistan: 14662.
PSF(PSF-NSFC/JSEP/ENG/AJKUKAJK/01)-NSFC: 12211540710.

### Competing Interests

Dragan Pamucar is an Academic Editor for PeerJ.

### Author Contributions

- Kamran Kausar performed the experiments, analyzed the data, prepared figures and/or tables, authored or reviewed drafts of the article, and approved the final draft.
- Khizar Hayat conceived and designed the experiments, performed the experiments, prepared figures and/or tables, and approved the final draft.
- Dragan Pamucar analyzed the data, prepared figures and/or tables, and approved the final draft.
- Nadeem Ajaib performed the computation work, authored or reviewed drafts of the article, and approved the final draft.

### Data Availability

The data is available in the tables.

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
