# Peer review of "Correlation coefficients on normal wiggly dual hesitant fuzzy sets: an application in the selection of real estate agents"

_PeerJ Computer Science, doi:10.7717/peerj-cs.3308_

## Round 0.1 · original submission · Major Revisions

· Academic Editor

Major Revisions

Reviewer 1 ·

Basic reporting

The goal of the article is to study the applicability of “normal dual wiggly hesitant fuzzy sets” to making choices for the selection of real estate agents. The paper includes a case study with its sensitivity analysis, plus a comparison with existing methods designed for related models.

This submission contains figures and tables that are readable, of enough quality, and descriptions to be published.

The structure is correct, and the article can be read and understood by specialists. English can benefit from further proofreading.

Experimental design

The article follows the standard steps in this literature:
- An original theoretical contribution, namely, correlation coefficients and their weighted counterpart, for the benchmark model.
- Description of the application to decision making (Sect 3), with a case study (Sect 4).
- Comparative analysis (Sect 5). It includes a sensitivity analysis.

Validity of the findings

The data are fully available, and the techniques are well described, hence, the readers can reproduce the findings. The conclusions are consistent with evidence and theoretical assumptions.

Additional comments

The paper can be accepted for publication. My minor suggestions follow.

First, I recommend a few minor changes to the Introduction:

1. The Introduction should not use abbreviations such as CC that are not common knowledge, even if they are defined in the Abstract

2. Line 38: Torra is the only author of [23].

3. The Introduction does not always achieve a progressive presentation of enhancements leading to the benchmark model. Consider lines 52-54. The transition from HFSs to NWHFSs (and then to its probabilistic version in line 59) is too abrupt, as the literature discusses other simpler enhancements. For example, the model defined in “Group consistency and group decision making under uncertain probabilistic hesitant fuzzy preference environment”, Inf. Sci. 414, 276–288 (2017), precedes the probabilistic dual hesitant fuzzy sets (DHFSs). DHFSs are mentioned later on the same page (line 63), and from this concept, we quickly move to NWDHFSs (line 73), however we need to know that other less sophisticated models such as “Dual extended hesitant fuzzy sets” (that were defined in Symmetry, 2019) are available too.

4. It is convenient to include an outline of the article at the end of the Introduction.

Other minor changes
I believe that the authors are not using the Definition environment correctly. See lines 91, 101, 103 (mid-line), 124, 133.

Line 174: There is a proof, but there is no Proposition or Theorem. I recommend putting the statement in lines 170-173 in a Proposition environment.

Normally, equation numbers are written in parentheses. Please check (lines 146, 170, 176, …).

Notation is very heavy. There are parts that would benefit from additional explanations and references. For example, what is the rationale for the many concepts in Eqs (8)-(15)? I believe they are related to the idea of score, which in the context of HFEs, is discussed extensively in a recent article that should be cited in lines 45-46 too: “Scores of hesitant fuzzy elements revisited: Was sind und was sollen”, Information Sciences 648 (2023), 119500

Reviewer 2 ·

Basic reporting

Motivation for the proposed article is missing in the introduction section.
The formatting of the article is not well designed.
Too many long equations and formulas are not understandable due to poor formatting.

Experimental design

Is the data used in the examples of this article real data? If so, explain it in a complete section.

Validity of the findings

Show the validity of the findings of the article.

Reviewer 3 ·

Basic reporting

The paper presents a solid idea and is generally well-written. I recommend minor revisions, as several areas still require improvement before it can be accepted for publication:
Abstract:
i. The abstract should be more comprehensive. The use of the word “we” should be minimized or avoided.
Introduction:
ii. The introduction largely focuses on the literature review. It is recommended to include more recent studies on MCDM methods such as https://doi.org/10.47852/bonviewJCCE2202366, https://doi.org/10.1155/2022/5172679.

Additionally, a subsection on motivation and novelty should be added to clearly explain the need and significance of this study.
Comparative analysis:
iii. In Section 5 (Comparative Analysis), the benefits of the proposed approach should be presented more clearly and in greater detail. Adding a dedicated subsection for this purpose would enhance clarity.
iv. The limitations of the proposed approach should be explicitly discussed. Also, the caption of Figure 2 is incorrect and needs to be revised.
v. The comparative analysis includes only two existing studies. Expanding the comparison to include more studies would strengthen the analysis.
Conclusion:
vi. The conclusion is missing a section number, and the discussion on future directions should be more detailed and insightful.

Experimental design

no comment

Validity of the findings

no comment

Additional comments

no comment

Reviewer 4 ·

Basic reporting

The manuscript is mainly clearly presented. However, the structure should be enhanced. Section 3 is too short and weak for a standalone section, and it lacks proper discussions of the results.

Experimental design

no comment

Validity of the findings

The conclusion and discussions of the results should be enhanced.

Additional comments

In general, this paper presents certain contributions to the relevant field. However, it requires significant improvements before considered for publication.
1. The writing style throughout the manuscript should be improved. The first word of the main text is an acronym, which is very uncommon.
2. The quality of the figures must be improved.
3. The motivations and novelties of this study are not appropriately described. The authors mentioned various extensions of fuzzy sets, however, the connections between these studies and this current study are not explained. The authors should properly define the motivations, research gaps and contributions of this study through the literature review.
4. The authors should distinguish preliminaries, methods and examples. The current preliminaries contain both research gap and examples, which should not be included in this section.
5. Section 3 is too weak. The authors should more extensively present and discuss the proposed method. Why is it better than other methods? What makes it unique?
6. The case study is weak. Is it practical for DMs to provide numerical DHFSs? In practice, the linguistic terms are more common. The authors should consider that.

---

## Round 0.2 · accepted · Accept

· Academic Editor

Accept

The reviewers seem satisfied with the content of this article's last version and therefore I can recommend this article for acceptance.

Reviewer 1 ·

Basic reporting

The goal of the article is to study the applicability of “normal dual wiggly hesitant fuzzy sets” to making choices for the selection of real estate agents. The paper includes a case study with its sensitivity analysis, plus a comparison with existing methods designed for related models.

This submission contains figures and tables that are readable, of enough quality, and descriptions to be published.

The structure is correct, and the article can be read and understood by specialists. Proofreading has improved readability.

Experimental design

The article follows the standard steps in this literature:
- An original theoretical contribution, namely, correlation coefficients and their weighted counterpart, for the benchmark model.
- Description of the application to decision making (Section 3), with a case study (Section 4).
- Comparative analysis (Sect 5). It includes a sensitivity analysis.

Validity of the findings

The data are fully available, and the techniques are well described; hence, the readers can reproduce the findings. The conclusions are consistent with evidence and theoretical assumptions.

Additional comments

Formatting is correct in the new version.

Reviewer 2 ·

Basic reporting

Motivation for the proposed research is appreciated.

Experimental design

Experimental designs are explained properly.

Validity of the findings

Validity of findings is clear.

Additional comments

The proposed idea in the article has potential for publication.